# Disruptive mutations in *TANC2* define a neurodevelopmental syndrome associated with psychiatric disorders

Hui Guo ⬛ et al.[#]

Postsynaptic density (PSD) proteins have been implicated in the pathophysiology of neuro-developmental and psychiatric disorders. Here, we present detailed clinical and genetic data for 20 patients with likely gene-disrupting mutations in *TANC2*—whose protein product interacts with multiple PSD proteins. Pediatric patients with disruptive mutations present with autism, intellectual disability, and delayed language and motor development. In addition to a variable degree of epilepsy and facial dysmorphism, we observe a pattern of more complex psychiatric dysfunction or behavioral problems in adult probands or carrier parents. Although this observation requires replication to establish statistical significance, it also suggests that mutations in this gene are associated with a variety of neuropsychiatric disorders consistent with its postsynaptic function. We find that *TANC2* is expressed broadly in the human developing brain, especially in excitatory neurons and glial cells, but shows a more restricted pattern in *Drosophila* glial cells where its disruption affects behavioral outcomes.

*email: alessandra.murgia@unipd.it; eee@gs.washington.edu   [#]A full list of authors and their affiliations appears at the end of the paper.

Neurodevelopmental disorders (NDDs) are a group of clinically and genetically heterogeneous conditions characterized by comorbidity of intellectual disability (ID), autism spectrum disorder (ASD), developmental delay (DD), language communication disorders, attention-deficit hyperactivity disorder (ADHD), motor abnormalities, and/or epilepsy[1]. NDDs show a wide range of overlapping clinical features, which pose serious challenges for diagnoses based exclusively on clinical criteria. Despite the exponential increase in the number of associated genes, the genetic basis for more than half of cases remains unknown, in large part, due to the rarity of particular genetic subtypes and more complex genetic patterns of inheritance (e.g., multifactorial).

Over the last decade, next-generation sequencing and microarray technologies have identified thousands of potentially pathogenic single-nucleotide variants (SNVs) or copy number variants (CNVs) and made it possible to expand the number of ASD/NDD high-risk genes into the hundreds. This research has benefited from the establishment of well-organized cohorts[2–6] and networks of investigators, such as the Simons Simplex Collection (SSC) cohort[7] and the Deciphering Developmental Disorders (DDD) network[8]. A recent integrated meta-analysis of whole-exome sequence (WES) data from 10,927 NDD families, for example, identified 253 NDD high-risk genes that showed an excess of likely gene-disrupting (LGD) or missense de novo mutations[9]. Nevertheless, the clinical significance of even those genes remains uncertain because in most cases the genes are represented by only a handful of cases often with limited clinical information.

The recent development of web-based platforms such as GeneMatcher[10] has facilitated international collaboration between clinical and research groups allowing individual genes to be explored in detail by coordinating the investigation of dozens of families and establishing detailed genotype–phenotype correlations[11–13]. Here, by prioritizing ASD candidate genes with subsequent targeted sequencing and international collaboration, we report genetic and clinical data for 20 individuals with disruptive mutations in *TANC2*, a gene encoding a synaptic scaffold protein interacting with multiple ASD and other neuropsychiatric disorder-related postsynaptic density (PSD) proteins in dendrites[14,15]. We define the NDD syndrome as attributed to the loss of *TANC2*; in addition, our data show that adult patients and carrier parents can present with a more complex series of psychiatric and behavioral disorders. The homolog of *TANC2* in *Drosophila*, *rols*, is expressed in larva and adult glial cells where its loss affects mating behavior raising the possibility that *TANC2* may also contribute to glial function.

## Results

**Prioritizing ASD candidate genes**. We reanalyzed WES data from 1902 SSC simplex quad families with matched unaffected siblings as controls[7]. We investigated whether there is significant LGD mutation burden in probands compared to unaffected siblings after excluding genes with genome-wide de novo significance in a recent large-scale meta-analysis[9]. After removing common LGD mutations (minor allele frequency (MAF) > 0.1% present in the Exome Aggregation Consortium (ExAC) nonpsychiatric subset)[16,17] and recurrent sites with low confidence, we annotated 307 de novo LGD mutations in probands (0.16 per individual) and 174 de novo LGD mutations in siblings (0.09 per individual) (Methods, Supplementary Data 1). As expected, probands showed a significant excess of de novo LGD mutations when compared to siblings ($P = 2.6 \times 10^{-9}$, ANOVA) (Methods and Fig. 1a). We repeated the analysis excluding variants in genes where genome-wide de novo significance had already been

established[9]. The filtered burden analysis revealed about one-third of the de novo LGD burden remained unaccounted (i.e., 0.13 LGD mutations per proband vs. 0.09 per sibling) ($P = 4.9 \times 10^{-4}$, ANOVA) (Fig. 1a).

Because ASD genes are significantly enriched as targets for FMRP and RBFOX binding[2,4], we further refined the filtered set of LGD candidates. We observed significant enrichment in FMRP ($P = 9.7 \times 10^{-9}$; OR = 2.2, Fisher's exact test) and RBFOX targets ($P = 2.2 \times 10^{-5}$; OR = 1.6, Fisher's exact test) (Fig. 1b) among probands but not siblings. We next assessed the distribution of each genome-wide significant gene's intolerance to mutation using the probability of being loss-of-function intolerant (pLI) score[16] and the residual variation intolerance score (RVIS)[18] as metrics (Supplementary Fig. 1). Based on these distributions, we excluded LGD mutations in genes with pLI score <0.84 and RVIS percentiles >32. Under such restrictions, the proband (1902 SSC probands) burden for de novo LGD mutations became more significant (0.040 vs. 0.019; $P = 2.5 \times 10^{-4}$, ANOVA) (Fig. 1a).

To prioritize ASD candidate genes, we combined de novo LGD mutations from two main ASD family-based WES studies: namely, the SSC WES study and the Autism Sequencing Consortium (ASC) WES study[4] (Methods). These two combined datasets represent 3953 families. Based on the SSC family analysis, we applied three criteria to prioritize the candidate genes: (i) mutation intolerance (pLI score > 0.84 and RVIS% < 32); (ii) FMRP and RBFOX target enrichment; and (iii) variants in genes where genome-wide de novo significance had not been established in the recent meta-analysis[9]. The procedure prioritized 58 ASD candidate genes for further consideration (Table 1, Fig. 1c, Supplementary Data 2).

**Targeted sequencing identified *TANC2* de novo mutations in ASD**. Using single-molecule molecular inversion probes (smMIPs)[19,20], we targeted 14 genes from the 58 candidate gene set for sequencing among 2154 Chinese ASD probands from the Autism Clinical and Genetic Resources in China (ACGC) cohort. We detected three LGD mutations: a de novo splice-site mutation in *TANC2* (NM_025185.3: c.1219+1G>A) and two maternally inherited variants in *SPTBN1*—a frameshift mutation (p.S8fs*8) and a splice-site variant (c.567-2A>C) (Supplementary Data 3). In addition, we analyzed all rare *TANC2* (ExAC nonpsychiatric subset MAF<1%) missense mutations ($n = 20$) and identified a de novo missense variant (c.2264 G > A; p.R755H). Combining the de novo mutations in *TANC2* identified in the SSC cohort (1 LGD and 1 missense), we observed a trend for an excess of LGD mutations in probands, although this observation did not remain significant after genome-wide multiple-test correction.

**Recruitment of a cohort of patients with *TANC2* mutations**. *TANC2* was recently reported to encode a synapse scaffolding protein interacting with multiple PSD proteins to regulate dendritic spines and excitatory synapse formation[14,15]. Our initial genetic findings and the strong functional implication warranted the investigation of more individuals with *TANC2* mutations. Using a network of international collaborators and GeneMatcher, a freely accessible website designed to enable connections between clinicians and researchers with a shared interest in a particular gene[10], we were able to identify an additional 17 families (17 probands and 3 affected siblings) with *TANC2* putative disruptive variants and a comorbid diagnosis of DD, ID, or ASD. These additional mutations included 14 LGD mutations and three intragenic microdeletions (Table 2, Supplementary Data 4). The variants were detected by WES, targeted sequencing, or array comparative genomic hybridization (aCGH) (Methods) in the

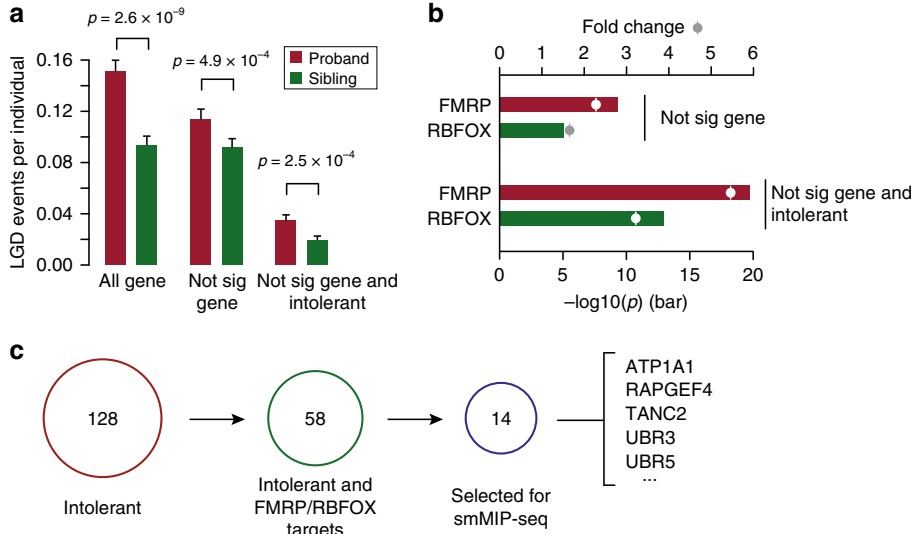

**Fig. 1** Prioritizing ASD candidate genes based on gene intolerance metrics and enrichment of FMRP/RBFOX targets. **a** Burden of de novo LGD mutations in probands of SSC simplex quad families for three categories: (i) all de novo LGD events; (ii) de novo LGD events excluding genes where significant burden has been reached; and (iii) de novo LGD mutations in intolerant genes without significance. The error bars represents a 95% confidence interval for the mean rates. Underlying data are provided as a Source Data file. **b** Enrichment of genes with de novo LGD mutations in FMRP and RBFOX targets. Enrichment is performed after excluding genes that reached significance (top panel) and the same but requiring that the genes are intolerant to mutation (bottom panel). **c** Selection of genes for targeted sequencing. One hundred and twenty-eight intolerant genes were prioritized by pLI score (<0.84) and RVIS percentile (>32) from the SSC and ASC cohorts, of which 58 genes were FMRP/RBFOX targets (Table 1). Fourteen genes were selected from the 58 genes for targeted sequencing in 2514 ASD probands from the ACGC cohort

corresponding collaborating centers either as part of routine diagnosis or as part of research studies.

In total, we identified and recruited 19 families (19 probands and 3 affected siblings) with potentially disruptive *TANC2* variants, including 16 LGD mutations and three intragenic microdeletions (Table 2, Fig. 2a, b). Among the 16 LGD mutations, we determined that 11 are de novo, 4 are inherited, and in 2 one parent's DNA was not available. Two of the three microdeletions are de novo and one is transmitted from the mosaic father to two affected siblings (Fig. 2b, c, Supplementary Fig. 2). *TANC2* is a highly conserved gene, in the top percentile of genes intolerant to truncating mutations (pLI score = 1, RVIS% = 0.4). No LGD mutations in *TANC2* are present in the 1000 Genomes Project[21], ESP6500 (ref. [22]), and 1911 SSC unaffected siblings. Only three LGD mutations in *TANC2* are observed in 45,376 ExAC nonpsychiatric samples, and none of the specific truncated mutations observed in our patients are present in this database.

Based on these data, we assessed significance of the genetic findings using two different tests. First, we assessed an enrichment of de novo LGD mutations among probands. Ten patients with confirmed *TANC2* de novo LGD mutations were screened from nine cohorts from a total of 16,113 tested individuals (ACGC, Amsterdam, BCM, Leiden, Leipzig, Melbourne, Paris, SSC, Toronto; Table 2). We estimate a significant excess for *TANC2* de novo LGD mutations compared with expected calculations ($P = 2.4 \times 10^{-14}$, chimpanzee–human divergence model (CH model); $P = 4.8 \times 10^{-14}$, denovolyzeR model, see Methods). This observation remained significant after genome-wide multiple-test correction ($P_{adj} = 9.2 \times 10^{-10}$, CH model; $P_{adj} = 1.9 \times 10^{-9}$, denovolyzeR model; Bonferroni correction for two models and ~19,000 genes). Because exome sequence coverage can differ among centers and affect the sensitivity of de novo mutation discovery, we gathered empirical data regarding the rate of de novo mutations among the different referring centers (Supplementary Table 1). Instead of using 1.5 as

the overall mutation rate in the CH model, for example, we repeated the analysis using 1.0 and 2.0 as extremes of the overall mutation rate in the CH model (Supplementary Table 1). The excess for *TANC2* de novo LGD mutations among probands remains significant under conditions of the most extreme de novo mutation rate (de novo mutation rate = 2.0, $P_{adj} = 7.7 \times 10^{-9}$; de novo mutation rate = 1.0, $P_{adj} = 8.4 \times 10^{-12}$) suggesting limited effect of differences in mutation detection sensitivity.

As a second test, we performed a more traditional burden analysis between cases and controls (Fisher's exact test). For example, we identified 13 probands with a known cohort size ($n = 17,567$). This represents a nominal enrichment of LGD mutations in NDD patients when compared to 45,375 ExAC nonpsychiatric controls ($P = 1.43 \times 10^{-5}$, OR = 11.2, Fisher's exact test, $P_{adj} = 0.29$, Bonferroni correction for 20,000 genes) (Methods). To guard against potential exon dropout, we assessed the mean coverage of *TANC2* exons in ExAC. The average coverage is 49.2 sequence reads per exon (Supplementary Fig. 3) with 24/25 of the exons showing on average more than 20-fold sequence read coverage.

**Disruption of *TANC2* defines an NDD syndrome.** Through patient recontact and review of the clinical information for all available patients/family members from the collaborating centers, we assembled phenotypic data for 19 probands and one affected sibling (13 males, 7 females, aged 4–40 years) carrying LGD mutations or intragenic microdeletions (Supplementary Data 4, Table 3). The most consistent phenotypes include ASD, ID, speech-language delay, and childhood motor delay (Table 3). In some cases, presentation of ASD was consistent with features reminiscent of the Rett-like spectrum. Of the 20 individuals assessed for ASD, seven cases had a formal ASD diagnosis, eight cases had clinical impression of ASD or Rett-like features, while the remaining five cases had no autistic phenotypes. Of the 20 individuals assessed for cognitive ability, 17 have a diagnosis of

**Table 1 Prioritized ASD candidate genes**

| Gene | ASD.LGD | Seq.Quality | MIS.sig | CLIN.sig | MIP-targets | FMRP/RBFOX | RVIS% | pLI score |
|------|---------|-------------|---------|----------|-------------|------------|-------|-----------|
| ATP1A1 | 1SP | validated | – | – | Yes | Both | 5.00 | 1.00 |
| CUX2 | 1SG | highConf | – | – | Yes | F | 9.20 | 1.00 |
| DSCAML1 | 1SP | validated | – | – | Yes | F | 0.67 | 1.00 |
| ELAVL2 | 1FS | validated | – | – | Yes | R | 15.85 | 0.96 |
| FAM91A1 | 1SG | validated | – | – | Yes | F | 21.65 | 0.88 |
| FBXW11 | 1SG | validated | – | – | Yes | R | 20.08 | 1.00 |
| RAPGEF4 | 1SP | highConf | – | – | Yes | Both | 9.59 | 1.00 |
| SMARCE1 | 1SP | highConf | – | Y(LGD) | Yes | R | 23.48 | 1.00 |
| SPTBN1 | 1SG | validated | – | – | Yes | F | 0.17 | 1.00 |
| TANC2 | 1SG | validated | – | – | Yes | Both | 0.37 | 1.00 |
| UBAP2L | 1FS | highConf | – | – | Yes | F | 3.33 | 1.00 |
| UBR3 | 1FS | validated | – | – | Yes | Both | – | 1.00 |
| UBR5 | 1FS | highConf | – | – | Yes | Both | 0.21 | 1.00 |
| ZNF462 | 1FS | highConf | – | Y(LGD) | Yes | F | 1.09 | 1.00 |
| BRSK2 | 1SP | highConf | – | – | – | Both | 4.95 | 0.89 |
| DST | 1FS | validated | – | – | – | Both | 0.28 | 1.00 |
| EP400 | 1FS | validated | – | – | – | Both | 7.32 | 1.00 |
| GRIN2B | 1FS,1SG,1SP | validated | Y | Y(LGD,MIS) | – | Both | 1.28 | 1.00 |
| NBEA | 1SG | validated | – | – | – | Both | 1.16 | 1.00 |
| NCKAP1 | 1FS,1SG | validated | – | – | – | Both | 3.85 | 1.00 |
| NRXN1 | 1SG | validated | – | – | – | Both | 1.78 | 1.00 |
| SKI | 1SG | highConf | – | – | – | Both | 11.79 | 0.97 |
| SPAG9 | 1FS | validated | – | – | – | Both | 14.41 | 1.00 |
| TRIM37 | 1SP | highConf | – | – | – | Both | 5.31 | 1.00 |
| BAI1 | 1SG | validated | – | – | – | F | – | 1.00 |
| BIRC6 | 1SG | highConf | – | – | – | F | 0.07 | 1.00 |
| CIC | 1SG | validated | – | Y(LGD) | – | F | 0.83 | 1.00 |
| DIP2A | 1SG,1FS | validated | – | – | – | F | 8.68 | 1.00 |
| DIP2C | 1FS | validated | – | – | – | F | 0.61 | 1.00 |
| DOT1L | 1SG | validated | – | – | – | F | 2.18 | 1.00 |
| KDM4B | 1SG | highConf | – | – | – | F | 13.00 | 1.00 |
| KIAA0100 | 1SG | validated | – | – | – | F | 2.12 | 0.92 |
| KIAA2018 | 1SG | highConf | – | – | – | F | – | 1.00 |
| NF1 | 1SG | validated | – | Y(LGD) | – | F | 0.39 | 1.00 |
| RALGAPB | 1FS | highConf | – | – | – | F | 3.00 | 1.00 |
| RELN | 1SG | highConf | – | – | – | F | 5.17 | 1.00 |
| SHANK2 | 1FS | validated | – | – | – | F | 2.03 | 1.00 |
| SMARCC2 | 1SP | validated | – | – | – | F | 8.51 | 1.00 |
| STXBP5 | 1FS | highConf | – | – | – | F | 4.70 | 1.00 |
| TRIO | 1FS | highConf | Y | – | – | F | 0.57 | 1.00 |
| BAZ2B | 1FS | validated | – | – | – | R | 13.09 | 1.00 |
| BRWD1 | 1FS | validated | – | – | – | R | 10.52 | 1.00 |
| CSDE1 | 1SG | validated | – | – | – | R | 6.18 | 1.00 |
| CUL1 | 1SP | highConf | – | – | – | R | 9.85 | 1.00 |
| ERBB2IP | 1SP | highConf | – | – | – | R | – | 1.00 |
| GABRB3 | 1FS | highConf | Y | | – | R | 25.36 | 1.00 |
| GGNBP2 | 1SG | highConf | – | – | – | R | 27.26 | 1.00 |
| GRIA2 | 1SG | highConf | – | – | – | R | 10.77 | 1.00 |
| HECTD1 | 1FS | validated | – | – | – | R | 0.45 | 1.00 |
| MPP6 | 1SP | validated | – | – | – | R | 26.90 | 0.99 |
| NFIA | 1SG | validated | – | – | – | R | 18.59 | 1.00 |
| NFIB | 1SP | validated | – | – | – | R | 16.62 | 0.98 |
| PCSK2 | 1FS | validated | – | – | – | R | 19.29 | 1.00 |
| PRPF40A | 1SP | highConf | – | – | – | R | 8.28 | 0.88 |
| RANBP2 | 1FS | highConf | – | – | – | R | 1.77 | 1.00 |
| UNC79 | 1FS | validated | – | – | – | R | 1.27 | 1.00 |
| XKR6 | 1SG | validated | – | – | – | R | 16.93 | 1.00 |
| YTHDC1 | 1FS | validated | – | – | – | R | 18.25 | 1.00 |

Notes: ASD.LGD represents LGD numbers and types in 3953 ASD patients from SSC and ASC cohorts. SG stopgain, FS frameshift, SP splice site. MIS.sig represents genome-wide significance for the de novo missense mutations in this gene based on the 10,927 NDD patients[9]. CLIN.sig represents whether there are clinical case-series reports for likely gene-disrupting (LGD) or missense (MIS) mutations of the specific genes

borderline to severe ID with two adult patients manifesting learning disability without formal ID diagnosis. Only one patient demonstrates an above-average IQ. Of the 20 individuals with language development records, 18 exhibited speech-language delay; at least 6 of them have absent language or severe language problems. Childhood motor delay was observed in 13 of 19 individuals for whom this information was available. In addition, 11/20 individuals had either a formal diagnosis of epilepsy ($n = 9$) or suffered from recurrent seizures ($n = 2$). Of the 15 patients with a constipation record, 8 reported chronic or severe

**Table 2 Summary of *TANC2* disruptive variants and de novo missense variants**

| Patient ID | Cohort | Cohort Size | Ascertainment | Methods | Function | NTchange | AAchange | Inheritance | ExAC[1] |
|---|---|---|---|---|---|---|---|---|---|
| **LGD mutations** | | | | | | | | | |
| HU1.p1 | BCM | 8910 | NDD | WES | Splice site | c.1219+1G>A | — | De novo | 0 |
| CC1.p1 | ACGC | 2154 | ASD | Target | Splice site | c.1219+1G>A | — | De novo | 0 |
| MA.p1 | Melbourne | 209 | EE | WES | Frameshift | c.1586_1587delAG | p.R530Kfs*5 | De novo | 0 |
| SS1.p1 | SSC | 2508 | ASD | WES | Stopgain | c.3196C>T | p.R1066* | De novo | 0 |
| TC.p1 | Toronto | 104 | DD/EE | WES | Frameshift | c.3828delA | p.E1277Kfs*7 | De novo | 0 |
| HU2.p1 | BCM | 8910 | NDD | WES | Splice site | c.4016+2T>G | — | De novo | 0 |
| AN.p1 | Amsterdam | 277 | ID/MCA | WES | Stopgain | c.4198C>T | p.Q1400* | De novo | 0 |
| LG.p1 | Leipzig | 100 | NDD | WES | Frameshift | c.4405delC | p.R1469Gfs*6 | De novo | 0 |
| NN1.p1 | Nijmegen | — | — | WES | Stopgain | c.4447C>T | p.Q1483* | De novo | 0 |
| LN.p1 | Leiden | 1200 | ID/DD | WES | Frameshift | c.4449delG | p.Q1483Hfs*69 | De novo | 0 |
| PF.p1 | Paris | 651 | DD | WES | Frameshift | c.5319_5344dup | p.F1782Cfs*6 | De novo | 0 |
| SS2.p1 | SSC | 2508 | ASD | WES | Splice site | c.547+1G>A | — | Paternal | 0 |
| NN2.p1 | Nijmegen | — | — | WES | Frameshift | c.2781delA | p.A928Qfs*4 | Maternal | 0 |
| TI.p1 | Troina | 1201 | DD | Target | Frameshift | c.2348_2349insCT | p.C784Sfs*22 | Paternal | 0 |
| GU.p1 | Greenwood | 253 | ID | Target | Splice site | c.3543+1G>T | — | Maternal | 0 |
| NN3.p1 | Nijmegen | — | — | WES | Frameshift | c.4713_4716delTCAG | p.Q1572Ffs*41 | Undetermined | 0 |
| **Microdeletions** | | | | | | | | | |
| PI.p1 | Padua | — | — | aCGH | Microdeletion | 240 kb deletion | — | De novo | — |
| DU.p1 | Davis | — | — | aCGH | Microdeletion | 146 kb deletion | — | De novo | — |
| CF.p1 | Caen | — | — | aCGH | Microdeletion | 194 kb deletion | — | Paternal | — |
| CF.p2 | Caen | — | — | aCGH | Microdeletion | 194 kb deletion | — | Paternal | — |
| DD.p1 | . | — | — | . | Microdeletion | 456 kb deletion | — | Undetermined | — |
| **De novo missense mutations** | | | | | | | | | |
| CC2.p1 | ACGC | 2154 | ASD | Target | Missense | c.2264G>A | p.R755H | De novo | 4 |
| NN4.p1 | Nijmegen | 100 | ID | WES | Missense | c.2278C>T | p.R760C | De novo | 0 |
| FS.p1 | SCZ | 623 | SCZ | WES | Missense | c.2381C>T | p.A794V | De novo | 4 |
| TU.p1 | TASC | 1045 | ASD | Target | Missense | c.2882G>A | p.R961Q | De novo | 0 |
| SS3.p1 | SSC | 2508 | ASD | WES | Missense | c.5066A>G | p.H1689R | De novo | 0 |

Notes: 1. Allelic numbers identified in 45,375 nonpsychiatric ExAC samples. CF.p1 and CF.p2 are affected siblings. CADD score of 4/5 de novo missense mutations are approximate or over 30; p.A794V, 34; p.R760C, 33; p.R755, 32; p.R961Q, 29.3. DD.p1 is from DECIPHER database. Isoform: NM_025185.3. Ascertainment represents the primary diagnosis of the corresponding cohorts. NDD neurodevelopmental disorders, ASD autism spectrum disorder, EE epilepsy, DD developmental delay, ID intellectual disability, MCA multiple congenital anomalies, SCZ schizophrenia

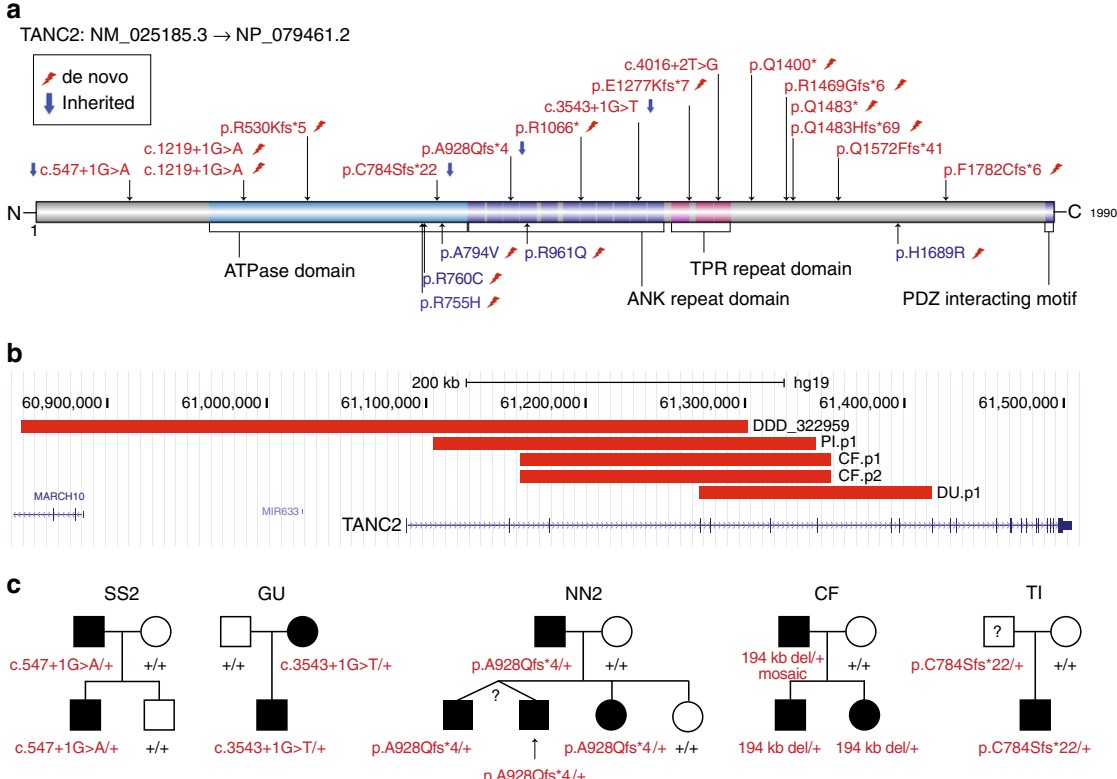

**Fig. 2** Location distribution and transmission pattern of *TANC2* mutations. **a** A protein domain graph (DOG) plot shows the positions of the 16 LGD (above) and 5 missense (bottom) mutations in *TANC2*. The annotated and predicted domains in *TANC2* are presented. A potential missense cluster is identified at the carboxy terminus of the ATPase domain. **b** Microdeletions identified in four patients (PI.p1, DU.p1, CF.p1, CF.p2) from this study and one patient (322959) in the DECIPHER database. CF.p1 and CF.p2 are affected siblings as noted in **c**. **c** Pedigree plot of the five families with transmitted disruptive variants. The carrier father in family SS2 has been diagnosed with behavioral and neuropsychiatric disorders, including bipolar disorder, ADHD, PTSD, and social issues reflected by the adult SRS score. The carrier mother in family GU has ID and experienced seizures, motor delay, and learning difficulties in school especially during her teenage years. The carrier father in family NN2 is suspected to have ID and has a psychiatric disorder history. The carrier father from the family CF experienced delayed motor development, learning difficulties in school, and also suspected to have ID. No clinical assessment of detailed developmental or neuropsychiatric history was possible for carrier father in family TI

constipation, 2 of which also presented with cyanotic extremities, indicating the common autonomic symptoms in this patient cohort.

Ataxia and skeletal anomalies were also observed in this patient cohort. Ataxia or spastic ataxia symptoms, for example, were observed in five patients; strabismus in five; progressive scoliosis or hyperkyphosis in five; foot deformities (club foot, flat valgus foot, talus valgus, inversion in ankles) in four; craniosynostosis (brachycephaly, turricephaly) in three; hypotonia in four; hypermobile joints in three; and chest deformities (pectus excavatum, pectus carinatum) in three (Table 3, Supplementary Data 4).

We observed some facial dysmorphic features with familial variability (Fig. 3), including large ears with thick helices, thick eyebrows with synophrys, deep-set eyes, strabismus, large nose with high nasal bridge, short and flat philtrum, large mouth with thin upper lip and thicker everted lower lip, widely spaced teeth (four patients), and tongue protrusion. A low hairline was commonly observed in patients of European descent.

**Transmission of *TANC2* mutations and psychiatric disorders.** Besides neurodevelopmental issues and association with ASD, we observed multiple significant psychiatric disorders or behaviors in the affected adult probands and carrier parents (Table 2, Supplementary Data 4). Three out of the six adult (21–40 years) probands showed different psychiatric problems: patient NN2.p1

presents with mood changes, aggressive behavior, and hyperactivity; patient NN3.p1 has hallucinations, compulsive behavior, and social-emotional delay; and patient SU.p1 suffers from major depression.

In four of the five families with transmitted variants, we observed mild neurodevelopmental phenotypes and/or psychiatric disorders in the carrier parents (Fig. 2c). In the simplex quad family SS2, the father had received a complex neuropsychiatric diagnosis, including bipolar, ADHD, and post-traumatic stress disorder. In addition, the father showed features suggestive of autism. For example, his SRS (Social Responsiveness Scale) falls in the top 8th percentile of all parents in the SSC (Supplementary Fig. 4). In this case, the father carried a splice-site mutation in *TANC2* (c.547+1G>A), which was transmitted to the affected proband but not the unaffected sibling. In family NN2, the carrier mother was also suspected of ID and had a history of undefined psychiatric disorder. The *TANC2* frameshift mutation (p. A928Qfs*4) was transmitted maternally to the proband and two affected siblings but not the unaffected sister. Interestingly, in family CF, the carrier father was determined to carry a low-grade somatic mosaicism for a 194 kb deletion (Chr17:61158866-61353248) and has suspected ID with a background of delayed motor development and school difficulties. In family GU, the carrier mother has ID and a history of seizures, motor delay, and learning difficulties during school years. Only the carrier father in family TI was thought to be unaffected; however, no clinical

**Table 3 Brief description of phenotypes of 20 probands or affected siblings with *TANC2* disruptive variants**

| Patient ID | SS2.p1 | CC1.p1 | HU1.p1 | MA.p1 | NN2.p1 | TI.p1 | SS1.p1 | GU.p1 | TC.p1 | HU2.p1 | AN.p1 | LG.p1 | NN1.p1 | LN.p1 | NN3.p1 | PF.p1 | DU.p1 | CF.p1 | CF.p2 | PI.p1 | Total |
|---|---|---|---|---|---|---|---|---|---|---|---|---|---|---|---|---|---|---|---|---|---|
| Mutation inheritance 13DN, 6INH | PI | DN | DN | DN | MI | PI | DN | MI | DN | DN | DN | DN | DN | DN | UD | DN | DN | PI | PI | DN | 13DN, 6INH |
| Age at last examination (years) | 12 | 5 | 23 | 21 | 31 | 15 | 15 | 12 | 12 | 6 | 7 | 14 | 7 | 4 | 40 | 15 | 4 | 22 | 27 | 16 | 6 Adult |
| Sex | M | M | F | M | M | M | M | M | M | F | M | M | M | F | F | F | M | F | M | F | 13 M, 7 F |
| **Neurodevelopmental problems** | | | | | | | | | | | | | | | | | | | | | |
| ASD/autistic features[1] | + | + | − | ± | + | − | + | ± | − | ± | ± | ± | + | + | + | − | − | ± | ± | ± | 15/20 |
| Intellectual disability[2] | − | + | ± | + | + | + | + | + | + | + | + | + | + | + | + | + | ± | + | + | + | 19/20 |
| Childhood speech delay | + | + | − | + | + | − | + | + | − | + | + | + | + | + | − | + | + | + | + | + | 18/20 |
| Childhood motor delay | − | − | − | + | | + | + | + | − | + | + | − | + | − | − | − | + | + | + | + | 13/19 |
| Regression | − | − | − | + | − | − | − | − | − | − | − | − | − | − | − | − | − | | | + | 2/12 |
| **Neurological problems** | | | | | | | | | | | | | | | | | | | | | |
| Epilepsy/seizure[3] | − | − | − | + | + | + | − | + | + | + | − | + | − | + | + | − | + | ± | ± | − | 11/20 |
| EEG abnormality | − | + | − | + | | + | | + | + | + | | + | + | + | + | − | + | − | − | − | 9/15 |
| Sleep disturbances | − | + | + | − | − | + | − | + | − | − | | + | + | | − | − | − | − | − | + | 5/13 |
| Microcephaly | − | − | − | − | | + | − | + | − | − | | + | − | | − | − | − | − | − | + | 3/18 |
| MRI brain abnormality | − | − | − | − | − | − | − | − | − | − | | + | | | | | | | | + | 2/16 |
| **Psychiatric and behavior problems** | | | | | | | | | | | | | | | | | | | | | |
| Repetitive behavior | + | + | − | + | + | − | + | + | + | − | + | + | + | + | + | − | − | + | − | + | 13/19 |
| Aggressive behavior | | − | + | − | + | + | − | − | − | + | | − | + | + | | | − | + | − | − | 6/16 |
| ADHD | + | + | + | − | + | − | | + | − | − | | − | − | + | | | − | − | − | + | 4/14 |
| Psychiatric problems | | − | + | − | + | − | | + | | | | + | | | + | | − | − | − | + | 3/15 |
| Anxiety | + | + | − | + | | − | − | − | − | − | | − | − | | | − | | + | + | | 4/12 |
| **Systemic problems** | | | | | | | | | | | | | | | | | | | | | |
| Chronic constipation | + | + | + | + | | + | | | | | | − | | + | | | | + | + | + | 9/15 |
| Inferior/spaced teeth | | − | − | + | | − | | + | | | | | + | | | | | + | + | + | 6/11 |
| Strabismus | | − | − | + | + | − | | + | − | | | | − | − | | | + | + | + | + | 6/14 |
| Ataxia/spastic ataxia | + | − | − | + | + | − | | + | − | | | + | − | − | | | + | + | + | + | 5/13 |
| Hypotonia | | − | − | + | | + | | − | − | | | | − | − | | − | + | + | − | + | 5/13 |
| Deformity of spinal column | + | − | − | + | | − | | + | | | | − | | | | − | − | + | + | + | 5/14 |
| Foot deformities | + | − | − | + | + | + | | + | − | − | | + | | | | − | | − | + | − | 5/15 |
| Chest deformities | | − | − | − | + | + | | + | | | | | | | | − | | | − | + | 3/12 |
| Craniosynostosis | | − | − | − | | − | | − | − | | | | − | | | − | − | | + | + | 3/14 |
| Hypermobile joints | | + | + | − | | + | | | | | | | | | | | | | | | 3/9 |

Notes: +, present; −, absent; blank, not reported. 1. +, ASD diagnosis, ±, Rett-like or autistic features; 2. +, borderline to severe ID, ±, learning disability without IQ test; 3. +, epilepsy, ±, seizure but no formal epilepsy diagnosis; CF.p1 and CF.p2 are affected siblings. Detailed clinical information is documented in Supplementary Table 4

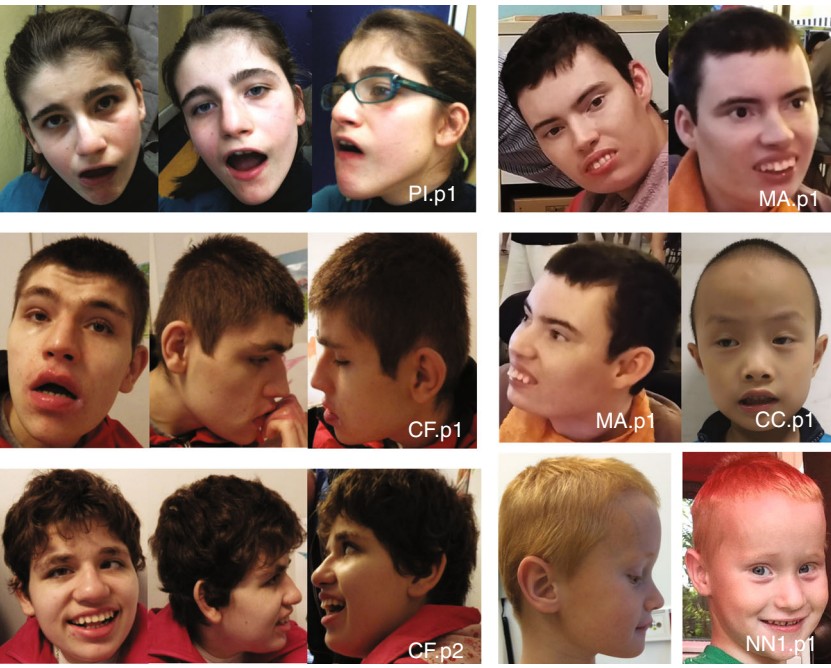

**Fig. 3** Facial dysmorphology. Proband photographs of PI.p1, CF.p1, CF.p2, MA.p1, NN1.p1, and CC.p1. Clinical similarities include low hairline (especially European-descent patients), large ears with thick helices, thick eyebrows with synophrys deep-set eyes, strabismus, large nose with high nasal bridge (for subset), short and flat philtrum, large mouth with thinner upper lip and thicker everted lower lip, inferior/widely spaced teeth (PI.p1, CF.p1, CF.p2, NN1.p1), and tendency for a protruded tongue. Consent for the publication of photographs was obtained for these patients

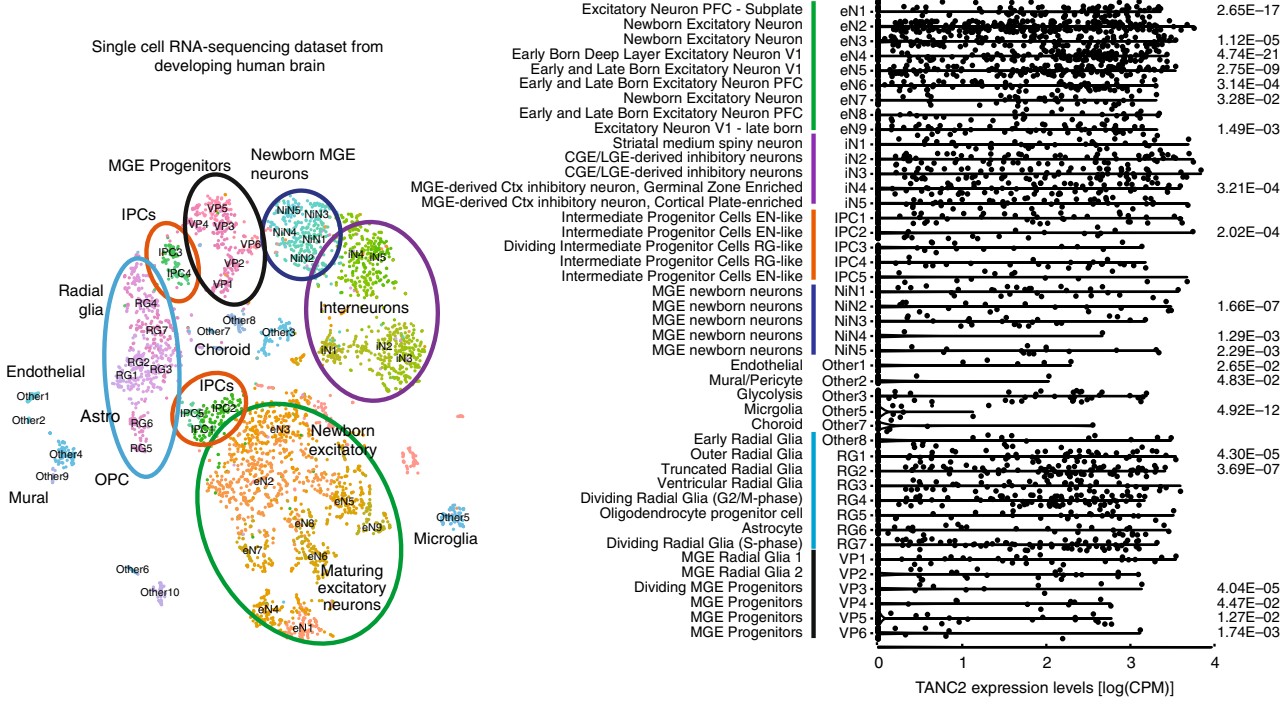

**Fig. 4** *TANC2* expression in human developing brain. **a** T-stochastic neighbor embedding of 4621 single-cell RNA sequencing (scRNA-seq) profiles from developing human brain samples identifies the major cell types in the developing brain. Cluster numbers are drawn directly from the source study, and biological interpretations can be found in Supplementary Table 3. **b** Violin plot showing *TANC2* expression across the major cell types identified in the scRNA-seq (**a**). Samples are ordered according to the average expression level across single cells of each type, and *p* value represents Bonferroni-corrected *p* value quantified using Wilcoxon rank sum test. n.s.—*p* > 0.05. Underlying data are provided as a Source Data file

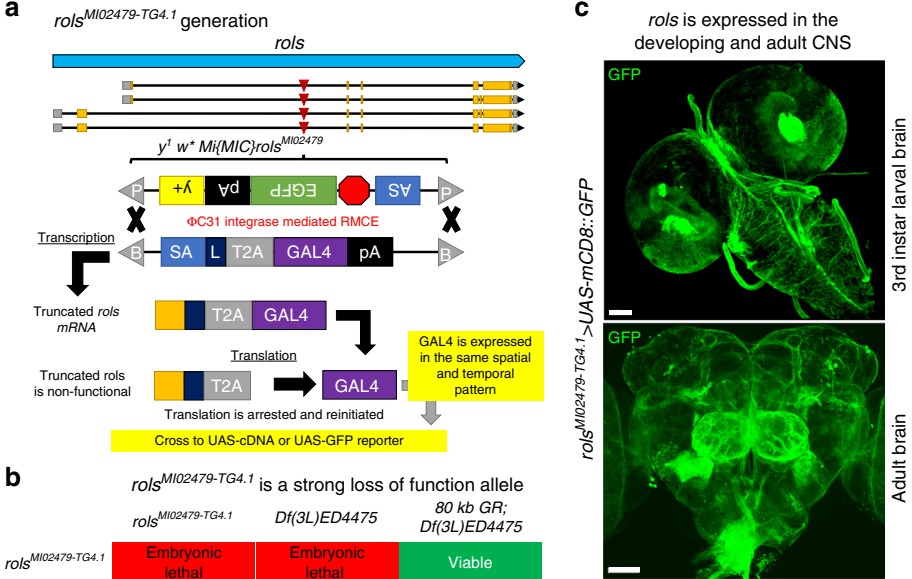

**Fig. 5** *rols* is an essential gene expressed in the developing and adult nervous system of flies. **a** Schematic of the *rols* locus where the *rols*[MI02479-TG4.1] allele was generated by genetic conversion of the *y*[1] *w** *Mi{MIC}rols*[MI02479] allele via RMCE using ΦC31 expression to swap the original *MiMIC* insertion cassette for an *SA-T2A-GAL4-polyA*. The *SA-T2A-GAL4* in the first intron of *rols* acts as an artificial exon resulting in early truncation of the *rols* transcript due to the polyA sequence. During translation, the T2A promotes ribosomal skipping and subsequent translation of GAL4 under the endogenous regulatory elements of *rols*. **b** Homozygous *rols*[MI02479-TG4.1] mutant flies are embryonic lethal. Embryonic lethality is also observed in *rols*[MI02479-TG4.1]/*Df(3L)ED4475*, which can by rescued by introduction of a 80 kb genomic rescue construct inserted on chromosome 2 (VK37). **c** Larval and adult brain staining (GFP) of *rols*[MI02479-TG4.1]>*UAS-mCD8::GFP* flies reveals membranous staining throughout the brain at both stages. Scale bar = 50 μm

follow-up or neuropsychiatric assessment could be performed. Combined, these observations indicate the important role of *TANC2* not only during neurodevelopment but also highlight the later risk of psychiatric or behavioral disorders.

**De novo missense variants in *TANC2*.** Although the most significant phenotypic findings were observed in patients with putative disruptive variants, we also identified five individuals with de novo *TANC2* missense mutations from denovo-db v.1.5 (ref. [23]) or by the targeted sequencing mentioned above (Table [2], Fig. [2]a, Supplementary Data 4). Three of these variants were identified in individuals with a primary diagnosis of ASD (p.R755H, p.R961Q, p.H1689R); one was found in an individual with a primary diagnosis of language disability and ID (p.R760C)[24]; and the last was identified in a patient with schizophrenia (p.A794V)[25], consistent with its role in psychiatric disorders. We note that three of the variants (p.R755H, p.R760C, p.A794V) cluster within the ATPase regulatory domain and one (p.R961Q) maps to one of the ANK domains (Fig. [2]a). Variants p.H1689R, p.R760C, and p.A794V have not been previously observed in ExAC nonpsychiatric samples while p.R755H and p.R961Q have both been reported four times in this database. Although the pathogenicity of these missense mutations is not yet established, p.R755H, p.R760C, and p.R961Q are predicted to have a damaging or possibly damaging effect (Supplementary Table 2) by multiple prediction tools. In addition, the specific p.R760C variant was recently reported to impair the recruitment of KIF1A-transported vesicles in neurons[14], making its pathogenicity more likely from a functional perspective.

**Expression pattern of *TANC2* in the developing human brain.** To further refine the expression pattern of *TANC2* in the developing human cerebral cortex, we analyzed a single-cell RNA sequencing dataset generated across 48 individuals[26]. Based on the transcriptomic profiles of 48 distinct clusters, we broadly classified six cell types: radial glia, intermediate progenitor cells, excitatory neurons, medial ganglionic eminence (MGE) progenitors, newborn MGE neurons, and inhibitory interneurons (Fig. [4]a, Supplementary Table 3). We find that *TANC2* is broadly expressed across many cell types, with enriched expression in excitatory neurons and radial glia, in particular truncated and outer radial glia (RG1 and RG2, Fig. [4]b, Supplementary Table 4). By contrast, MGE-derived newborn interneurons and their progenitors in the medial ganglionic eminence are generally depleted for *TANC2* expression. Using the same approach, we find that *TANC1*, the paralogs of *TANC2*, shows an even stronger pattern of enriched expression in radial glial cells (Supplementary Fig. 5).

**Disruption of *TANC2* (*rols*) interferes with synapse function.** The closest *Drosophila melanogaster* homolog to human *TANC2* is *rols*, which is highly conserved at the amino acid level (55% similarity, 38% identity). Rols is the only TANC-like protein in *Drosophila* with a DIOPT[27] score of 11/15 for *TANC2* and 9/15 for *TANC1*, making it the sole fly homolog to both human TANC proteins[28]. To investigate whether *TANC2* and *Drosophila rols* function in a similar manner, and whether its disruption influences neurodevelopment in flies, we generated new *rols* alleles and carried out a series of expression and phenotype analyses along with tissue-specific RNAi-mediated knockdown and rescue experiments.

To investigate the expression pattern of *rols* in *Drosophila*, we generated *rols*[MI02479-TG4.1] by genetic conversion of the *rols*[MI02479] allele via recombination-mediated cassette exchange as previously described[29–31] (Fig. [5]a). Insertion of an SA (splice acceptor)-T2A-GAL4-polyadenylation (polyA) signal creates an artificial exon between exons 1 and 2 of *rols*, predicted to cause premature transcriptional arrest as well as expression of GAL4 under the endogenous regulatory elements of *rols*. As expected, *rols*[MI02479-TG4.1] mutants are embryonic lethal and are also lethal at this stage when *in trans* with a corresponding deficiency

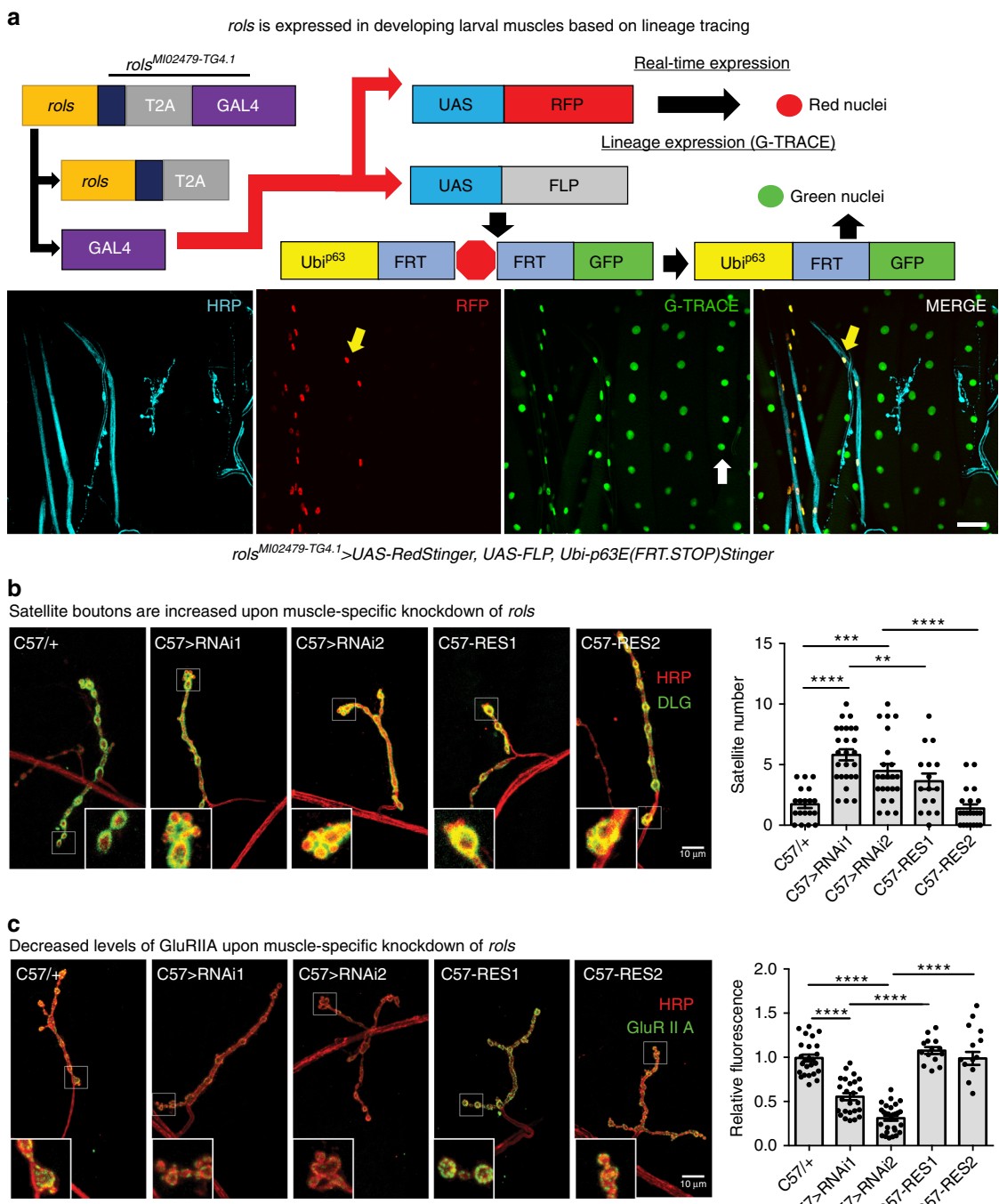

**a** *rols* is expressed in developing larval muscles based on lineage tracing

*rols^(MI02479-TG4.1)>UAS-RedStinger, UAS-FLP, Ubi-p63E(FRT.STOP)Stinger*

**b** Satellite boutons are increased upon muscle-specific knockdown of *rols*

**c** Decreased levels of GluRIIA upon muscle-specific knockdown of *rols*

**Fig. 6** *rols* is historically expressed in developing muscle cells and muscle-specific knockdown of *rols* increases satellite bouton number and decreases GluRIIA levels at the third instar larval NMJ. **a** G-TRACE analysis by crossing *rols^(MI02479-TG4.1)* to *UAS-RedStinger, UAS-FLP, Ubi-p63E(FRT.STOP)Stinger* reveals that Rols is currently expressed in wrapping glia at the NMJ (yellow arrow) and was historically expressed in muscle cells (white arrows). Scale bar = 50 μm. **b, c** Muscle-specific (*C57-Gal4*) knockdown of *rols* showed an increased number of satellite boutons (**b**) (C57/+, $n = 19$; C57>RNAi1, $n = 26$; C57>RNAi2, $n = 23$; C57-RES1, $n = 16$; C57-RES2 $n = 21$) and decreased normalized fluorescent intensity of GluRIIA (**c**) (C57/+, $n = 25$; C57>RNAi1, $n = 25$; C57>RNAi2, $n = 28$; C57-RES1, $n = 14$; C57-RES2 $n = 13$) compared to wild-type larvae. Scale bar = 10 μm. Underlying data are provided as a Source Data file. Statistical results for **b** and **c**. ****$p < 0.0001$, ***$p < 0.001$, **$p < 0.01$, *$p < 0.05$; n.s. not significant. Error bars represent SEM

(*Df(3L)ED4475*). Expression of human *UAS-TANC2* failed to rescue lethality but viability was restored upon introduction of an 80 kb P[acman] genomic BAC rescue (GR) construct (Fig. 5b)[32]. These data indicate *rols^(MI02479-TG4.1)* is a loss-of-function allele, and that this allele is responsible for the lethality in the flies.

To determine if *rols* is expressed in the larval or adult brain we used *rols^(MI02479-TG4.1)* to drive *UAS-mCD8::GFP* revealing the membranes of cells expressing *rols* (Fig. 5c). The Rols protein is clearly expressed throughout the nervous system. Interestingly, the *rols* expression pattern is similar to other genes previously documented in glia[31]. We also generated an internally GFP-tagged protein trap (*rols^(MI02479-GFSTF.1)*) allele[33]; however, we were unable to detect endogenous Rols protein in vivo. In our experience, this result is not uncommon, particularly for proteins

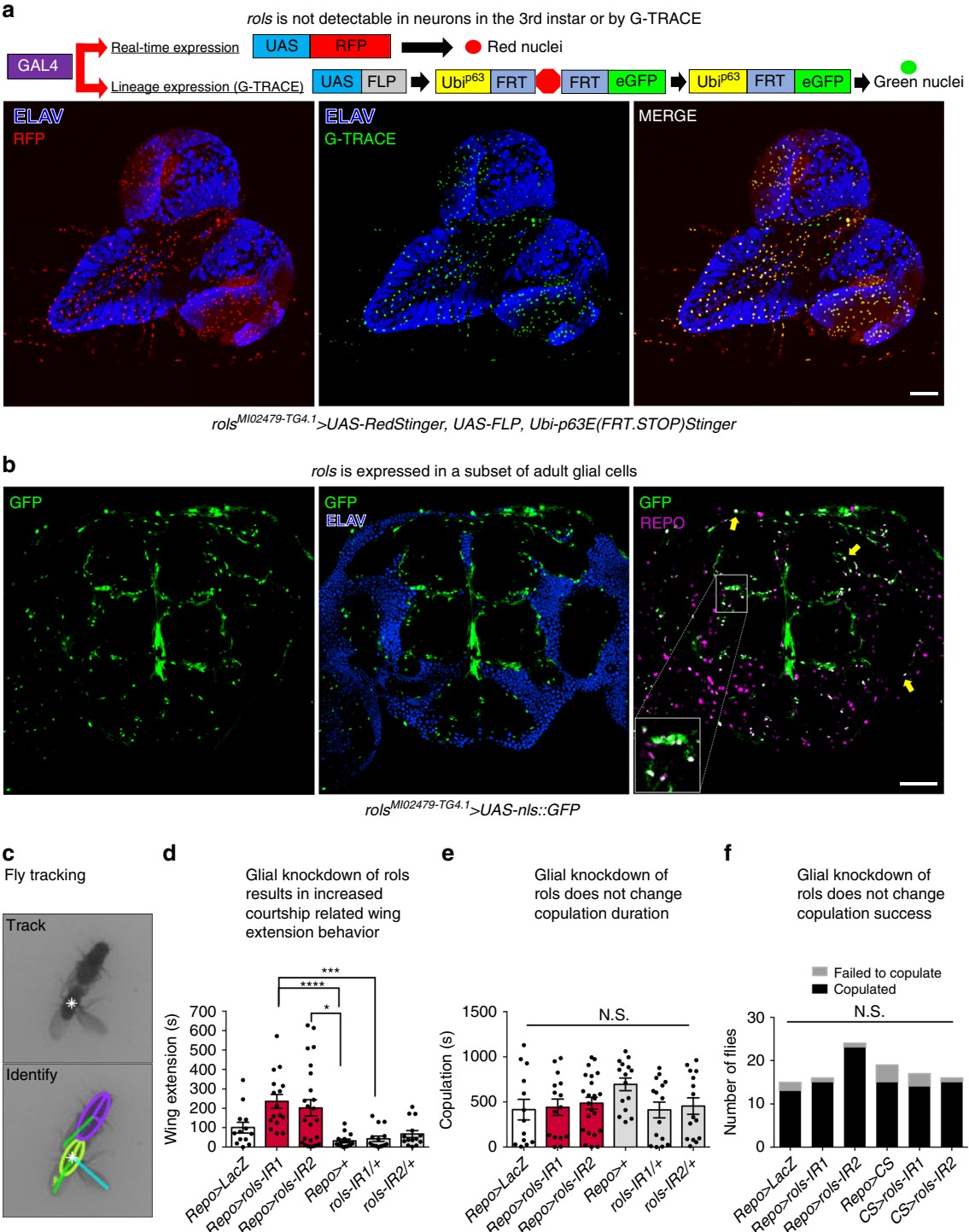

**Fig. 7** *rols* is enriched in glia and glial-specific knockdown of *rols* leads to increased courtship display frequency in adult flies. **a** Pan-neuronal staining (ELAV) of *rols*[MI02479-TG4.1]>*UAS-RedStinger, UAS-FLP, Ubi-p63E(FRT.STOP)Stinger* third instar larva reveals that *rols* is not expressed in neurons. Scale bar = 100 μm. **b** Neuronal (ELAV) and glial (REPO) staining of adult fly brains using *rols*[MI02479-TG4.1] > *UAS-nls::GFP* reveals that *rols* is expressed in a subset of glia (yellow arrows, and inset on right panel). Scale bar = 100 μm. **c–f** Glial-specific (*Repo-Gal4*) knockdown of *rols* (IR1, $n = 16$; IR2, $n = 24$) leads to increased courtship display duration when compared to controls (LacZ, $n = 13$; +, $n = 15$; IR1/+, $n = 15$; IR2/+, $n = 14$) (**c, d**) while copulation duration and success remained unchanged (**e, f**). Statistical results for **d–f**. ****$p < 0.0001$, ***$p < 0.001$, *$p < 0.05$; n.s. not significant. Error bars represent SEM. Underlying data are provided as a Source Data file

that are low abundance. Hence, we focused on cell-type expression of *rols* using the *rols*[MI02479-TG4.1] allele. To determine the cell types that express *rols* during development, we utilized Gv-TRACE[34]. This labels the cells that expressed *rols* previously as well as the cells that currently express *rols*. Using HRP to stain neuronal axons, we determined that *rols* is not actively expressed in the muscles of third instar larvae (Fig. 6a, RFP positive nuclei) but was present in muscle cells earlier in development (Fig. 6a,

GFP positive nuclei). We also observed *rols* expression in the nuclei of wrapping glia at the neuro-muscular junction (NMJ) (Fig. 6a, yellow arrows).

Since TANC2 is a PSD-interacting protein and we observed lineage expression of *rols* in the muscles, we investigated whether reduction of *rols* influences synapse growth and function in postsynaptic membranes. Using a muscle-specific driver (C57-GAL4), we knocked-down *rols* in muscles via RNAi and examined the third instar NMJ. We co-labeled the NMJs with axonal (HRP) and postsynaptic (DLG) markers. Morphological analysis was carried out for type Ib boutons at muscle 4 of abdominal segment 3/4. Compared to wild type, we observed a significant increase in mean satellite bouton number with *rols* knockdown, and this overgrowth was rescued by expression of UAS-rols in muscle (Fig. 6b). To determine whether *rols* knockdown interferes with the expression or localization of postsynaptic molecules, we examined the distribution of glutamate receptor IIA (GluRIIA) subunit levels in type Ib boutons at muscle 4 of abdominal segment 3/4. For quantification of wild type and knockdown lines, we measured synaptic GluRIIA immunofluorescence intensity normalized to wild type. We observed a significant reduction of the synaptic immunoreactivity of GluRIIA in the knockdown lines. This defect was also rescued by expression of *rols* in muscle (Fig. 6b). These data indicate that Rols in the postsynapse is important for proper synaptic morphology and glutamate receptor clustering the normal development of the NMJ.

We were unable to detect *rols* in the vast majority of *Drosophila* neurons, including motor neurons. Lineage tracing showed no evidence of historical *rols* expression in neurons (Fig. 7a). Instead, we observed expression almost exclusively in glia (Fig. 7a, non-ELAV+ cells). Thus, we examined the adult fly brain of *rols*[MI02479-TG4.1]>UAS-nls::GFP flies stained for neuronal (Elav) and glial (Repo) markers. Consistently, *rols* is expressed in a subset of glia and very few neurons also express *rols* (Fig. 7b). Interestingly, the glial-specific expression of *rols* in the adult brain is supported by recent single-cell transcriptomic data[35].

Based on these data, we conducted glial-specific (repo-GAL4) knockdown of *rols* and assessed adult flies for changes in behavior. We did not detect any climbing or stress-induced seizure-like behavior with glial or neuronal knockdown (Supplementary Fig. 6). Hence, we examined *Drosophila* courtship behavior to determine if there were detectable changes in neurological function. In *Drosophila*, courtship is a stereotyped social behavior that males are capable of performing without prior training or experience. It is composed of a series of sequential steps required for the initiation of copulation. One of these steps, the production of courtship song through wing vibration persuades the female to copulate and requires the coordination of multiple sensory and motor systems. We isolated male pupae before eclosion to prevent the possibility of experience-based change in behavior. Isolated flies were inspected, aged for 3–5 days, and placed in the test chamber with a single wild-type female. All interactions were filmed for a 30-min period, and the video was analyzed using automated tracking and behavior detection software[36,37] (Fig. 7c). Using this method, we observed that repo-GAL4>UAS-rols-RNAi flies displayed a significant increased total duration of wing-extension behavior while maintaining wild-type levels of locomotion, grooming, and copulation (Fig. 7d–f, Supplementary Fig. 7). Importantly, pan-neuronal knockdown of *rols* using elav-GAL4 did not alter wing-extension behavior (Supplementary Fig. 8).

In summary, using *Drosophila*, we confirmed the evolutionarily conserved role of this gene in the postsynapse and found roles in glia that modulate behavior.

## Discussion

This study reports the phenotypic spectrum associated with severe or disruptive TANC2 mutations for 20 probands or affected siblings from 19 unrelated families. The neurodevelopmental phenotype is characterized by ASD, ID, speech and motor delay, and facial dysmorphology. Other clinical features include epilepsy, autonomic dysfunction, movement alterations, including stereotypies and altered gait, and signs of connective tissue abnormalities. Features of this disorder are reminiscent of Rett-like phenotypes, at least for some of the most severely affected.

Several lines of evidence reinforce that the mutations in TANC2 play an important role in NDDs. First, we identify a total of 16 TANC2 truncating variants, including ten de novo SNVs and two de novo microdeletions. Calculation of the enrichment of ten de novo LGD mutations with available cohort sizes reaches genome-wide significance for an excess of de novo mutation. Second, we provide a consistent phenotypic profile of the 19 probands and one affected sibling recruited in this international study. Third, we observe mild neurodevelopmental impairment or neuropsychiatric disorders in carrier parents in four out of five families with transmitted disruptive TANC2 variants or microdeletions. Finally, a recent functional study showed that proteins with a disease-related LGD mutation (p.R1066*) failed to accumulate at the dendritic spines[14], indicating a critical role of the predicted C-terminal PDZ interacting motif[38] in disease pathogenesis. All disruptive mutations identified in our patient cohort delete or truncate before the predicted PDZ interacting motif.

TANC2 is a synaptic scaffold protein that interacts and co-localizes with PSD proteins in dendrites in various brain regions[15]. During rat brain development, Tanc2 expression is detectable at embryonic stages and persists postnatally, albeit with reduced expression[15]. Disruption of Tanc2 positively influences dendritic spines and excitatory synapse formation[14,15]. In mouse models, the Tanc2 knockout associates with embryonic lethality consistent with a critical role in early-stage embryonic development[15]. Although the mechanisms of TANC2 function remain to be investigated, known protein interactions include the products of numerous ASD and neuropsychiatric disorder risk genes (e.g., PSD95, SHANK1, SYNGAP1, CASK, and GRIN2B)[14]. Deciphering the function of TANC2 will undoubtedly provide additional insights with respect to PSD-related pathogenesis and its importance with respect to converging NDDs and psychiatric disorders.

In *Drosophila*, a reduction of *rols* in muscles results in the reduction of postsynaptic GluRIIA levels. However, we only detected *rols* expression in muscle early in development and not throughout larval developmental stages. The effects of *rols* knockdown in embryonic muscle using C57-GAL4 may affect NMJ development or the perdurance of Rols in larval muscles may affect the organization of the PSD protein and cause a decrease in GluRIIA levels when Rols is lost progressively. We also observed an increase in presynaptic satellite bouton number upon reduction of *rols* in muscle. Typically, satellite bouton formation is determined by presynaptic processes usually associated with defects in synaptic vesicle endocytosis that lead to an increase in satellite boutons[39–41]. In contrast, knockdown of the postsynaptic protein Diablo results in an increase in satellite bouton number[42]. The latter is correlated with an increase in GluRIIA levels. Hence, the mechanisms underlying these NMJ defects when *rols* is affected remain to be determined.

In humans, TANC2 is broadly expressed in different cell types of the developing brain but shows enrichments in both excitatory neurons and radial glial cells, which act as neural stem cells. In contrast, we find that *rols* expression is restricted mainly to glia cell types in larvae and adult flies. Reduction of *rols* in glia increased the total amount of time *Drosophila* males spent

performing courtship-related wing extensions. In many animals, courtship is a complicated neurological process that involves integrating sensory input, multiple components of the central nervous system and motor output. In *Drosophila*, it has previously been shown that alterations in glial function are sufficient to alter courtship behavior, demonstrating that changes in glia can have consequences on behavioral output[43]. Our results suggest that *rols* in supporting glia may act in modulating neuronal function. Intriguingly, TANC2 as well as other PSD-related proteins are also expressed in radial glia[44], which act as neural stem cells and give rise to both excitatory neurons and macroglia. Moreover, the second human homolog of *rols*, *TANC1*, shows a more restricted expression during development, with enriched abundance in radial glia, astrocytes, and newborn neurons, suggesting the possibility of subfunctionalization[45] as well a possible neofunctionalization of ancestral *rols* function during vertebrate evolution. Our results suggest that the functional consequences of NDD gene variants in neuron–glia communication should be investigated in vertebrate models to assess if they may underlie the clinical heterogeneity of NDD.

In summary, we have demonstrated an excess of de novo disruptive *TANC2* mutations among patients with neurodevelopmental delay and autism. Our results suggest that *TANC2* is one of a growing list of genes where mutations associate with both pediatric neurodevelopmental and adult neuropsychiatric disease, for example, *SHANK3* (ref. [46]), *NRXN1* (ref. [47]), *ZMYND11* (ref. [48]), *POGZ*[25], and *RELN*[49]. Because disruptive mutations may also be inherited, much larger cohorts of both cases and controls will be needed to establish its involvement with neurodevelopment and psychiatric disease. As such, the detection of pathogenic *TANC2* mutations will not only lead to a causal diagnosis for patients in such populations but also be important for a better understanding of genotype–phenotype associations and in guiding precision medicine. Animal models and deeper molecular functional studies on the cell-specific role of *TANC2* in neurodevelopment will not only provide insights regarding PSD pathogenesis in ASD and other neuropsychiatric disorders but also provide an avenue for the development of long-term treatment of these disorders.

## Methods

**Prioritizing ASD candidate genes**. De novo LGD mutations were analyzed from two ASD WES cohorts: (1) SSC[2], which included 2508 ASD probands and 1911 unaffected siblings from 615 trios and 1902 simplex quad families; and (2) ASC [4], which included 1445 ASD probands from trio families. All de novo LGD mutations collected from the two cohorts were re-annotated with ANNOVAR (2018Apr16)[50]. Conflicts between ANNOVAR annotation and the original annotation were manually curated. We removed variants that were observed in the general population (MAF > 0.1% in ExAC nonpsychiatric subset) and LGD sites with low quality. Thresholds of mutation intolerance (pLI score and RVIS) were determined according to the distribution of pLI score and RVIS percentage of genes with significance (FDR corrected $q < 0.05$) in Coe et al.[9] (Supplementary Fig. 1). Enrichment of de novo LGD mutations between probands and siblings from 1902 SSC simplex quad families was performed using ANOVA considering paternal birth age as a covariate. Enrichment analysis of genes with de novo LGD mutations in probands and siblings in 842 FMRP targets (with transcripts bound by the fragile X mental retardation protein)[51] and 1488 RBFOX targets (merge of 1048 binding targets and 578 splicing targets)[52] was performed using Fisher's exact test using all protein-coding genes expressed in brain as the background[53].

**ACGC cohort and targeted sequencing**. Targeted sequencing of the 14 candidate genes was performed on 2154 ASD probands of complete parent–child trios or quads from the ACGC cohort, which has been described previously[20]. In brief, ACGC patients are diagnosed primarily according to DSM-IV/V criteria documenting additional comorbid conditions where possible. Peripheral-blood DNA of all probands, parents and siblings, where available, was collected with informed consent. Genomic DNA was extracted from the whole blood using a standard proteinase K digestion and phenol–chloroform method. The sequencing data have been deposited in the National Database for Autism Research (NDAR)[54]. The study complied with all relevant ethical regulations for work with human participants and was approved by the Human Ethics Committee of Center for Medical Genetics, Central South University (institutional review board (IRB) #2014031113).

Targeted sequencing was performed using smMIPs[19,20]. In summary, smMIPs were designed using MIPgen with an updated scoring algorithm. After amplification, libraries were sequenced using the Illumina HiSeq2000 platform. Sequences were aligned against GRCh37 using BWA-MEM (v.0.7.13)[55] after removing incorrect read pairs and low-quality reads. SNVs/indels were called with FreeBayes (v.0.9.14)[56]. Variants exceeding tenfold sequence coverage and read quality over 20 (QUAL > 20) were annotated with ANNOVAR using reference GRCh37. LGD variants and rare missense variants (MAF < 0.1% in ExAC nonpsychiatric subset) were selected for validation using Sanger dideoxy sequencing.

**Statistical analysis for excess de novo LGD mutations**. The probability of de novo LGD mutation enrichment was calculated using two statistical models: (1) a binominal model that incorporates gene-specific mutation rates estimated from the overall rate of mutation in coding sequences, estimates of relative locus-specific rates based on CH model[19] with an expected rate of 1.5 de novo mutations per exome, and (2) a Poisson model that uses gene-specific mutation rates estimated from trinucleotide context and accommodates known mutational biases, such as CpG hotspots[57,58]. *TANC2* mutation burden between the disorder cohort and 45,375 ExAC nonpsychiatric samples was performed using Fisher's exact test. All statistical analyses were performed using the statistical software R (v3.2.1) (www.r-project.org/).

**Patients and assessment**. The probands carrying *TANC2* LGD mutations or microdeletions and their family members were recruited to different participating centers from eight countries. For each affected individual, detailed clinical information was obtained through patient recontact or detailed review of medical records by neurologists, pediatricians, or geneticists. Written informed consent was obtained from study participants or their parents or legal guardians in the case of minors or those with ID, in line with local IRB requirements at the time of collection. Genomic DNA was extracted from the whole blood of the affected individuals and their parents. Parents and affected/unaffected siblings of the probands where available were also recruited for segregation analysis and phenotyping. The authors affirm that human research participants provided informed consent for publication of the images in Fig. 3.

**Variant detection**. The 20 potentially disruptive variants in *TANC2* were identified by WES, targeted sequencing, or aCGH following the standard guidelines at the participating centers. The details are described below.

WES was applied for detection of variants in individuals NN1.p1, NN2.p1, NN3.p1, NN4.p1, AN.p1, LN.p1, PF.p1, TC.p1, LG.p1, SS1.p1, SS2.p1, HU1.p1, and HU2.p1. The LGD mutations in patients SS1.p1 and SS2.p1 were detected by WES from the SSC cohort[2,59].

The LGD mutations in patients HU1.p1 and HU2.p1 were detected using clinical WES at the Human Genome Sequencing Center at Baylor College of Medicine. The WES protocol, including library construction, exome capture, and HiSeq next-generation sequencing and data analysis have been described elsewhere[60].

The LGD mutations in patients NN1.p1, NN2.p1, and NN3.p1 and the de novo missense mutation in NN4.p1 were detected by WES at the Nijmegen Radboud University Hospital and at the Maastricht University Medical Center. Routine diagnostic exome sequencing and variant calling using a parent–offspring trio approach was performed[24]. Briefly, the exome was captured using the Agilent SureSelectXT Human All Exon v5 library prep kit (Agilent Technologies, Santa Clara, CA, USA). Exome libraries were sequenced on an Illumina HiSeq 4000 instrument (Illumina, San Diego, CA, USA) with 101 bp paired-end reads at a median coverage of 75× at the BGI Europe facilities (BGI, Copenhagen, Denmark). Sequence reads were aligned to the hg19 reference genome using Burrows-Wheeler Alignment (BWA) version 0.5.9-r16.14[55]. Variants were subsequently called by the Genome Analysis Toolkit (GATK) unified genotyper, version 3.2-2 and annotated using a custom-built diagnostic annotation pipeline.

The LGD mutation in patient LN.p1 was detected by WES at Department of Clinical Genetics, Leiden University Medical Center, Leiden, The Netherlands. Exomes were captured using the Clinical Research Exome v2 capture library kit (Agilent, Santa Clara, CA, USA) accompanied by Illumina paired-end sequencing on the NextSeq 500 (Illumina, San Diego, CA, USA). The in-house sequence analysis pipeline Modular GATK-Based Variant Calling Pipeline (MAGPIE) (LUMC Sequencing Analysis Support Core, LUMC) was used to call the SNVs/indels. An in-house developed annotation pipeline was used to annotate the variants.

The LGD mutation in patient PF.p1 was detected by WES at the Département de Génétique, Hôpital Pitié-Salpêtrière, Paris, France. DNA was extracted from maternal, paternal, and proband samples. Trio WES was performed on a NextSeq 500 Sequencing System (Illumina, San Diego, CA, USA), with a 2 × 150 bp high-output sequencing kit after a 12-plex enrichment with SeqCap EZ MedExome kit (Roche, Basel, Switzerland), according to the manufacturer's specifications.

Sequence quality was assessed with FastQC 0.11.5, reads were then mapped using BWA-MEM (v.0.7.13), sorted, and indexed in a BAM file (SAMtools 1.4.1), duplicates were flagged (sambamba 0.6.6), and coverage was calculated (Picard-tools 2.10.10). Variant calling was done with GATK 3.7 Haplotype Caller[61]. Variants were then annotated with SnpEff 4.3, dbNSFP 2.9.3, gnomAD, ClinVar, HGMD, The Human Variome Project Great Middle East, and an internal database. Coverage for these patients was 93% at a 20× depth threshold.

The LGD mutation in patient TC.p1 was detected by WES using genomic DNA from the proband and parents in a clinical molecular genetics laboratory, GeneDx; the exonic regions and flanking splice junctions of the genome were captured using the Clinical Research Exome Kit (Agilent Technologies, Santa Clara, CA, USA). Short-read sequencing was performed using the Illumina platform with ~100 bp paired-end reads. Sequence variants were called based on alignment to the human reference genome, GRCh37/UCSC hg19, as previously described[62].

The LGD mutation in patient LG.p1 was detected by WES at University of Leipzig Hospitals and Clinics, Leipzig, Germany. Trio WES for the proband and his biological parents was performed. DNA was subjected to exome capture using Agilent SureSelectXT library preparation and human all exon capture (V6; Agilent, Santa Clara, CA, USA). Sequencing was performed on an Illumina NovaSeq6000 (2 × 100 bp, Illumina, San Diego, CA, USA) by a commercial provider (CeGaT, Tübingen, Germany) on our request. Bioinformatics processing and filtering was performed using the software Varfeed and Varvis (Limbus, Rostock, Germany). This study was approved by the ethics committees of the University of Leipzig (402/16-ek).

The LGD mutation in patient MA.p1 was detected by WES at University of Washington Center for Mendelian Genomics (UW-CMG, Seattle, WA, USA). Library construction and WES were performed for the proband–parents trio. The exome was captured using the NimbleGen SeqCap EZ Exome 2 (Roche NimbleGen, Madison, WI, USA). Exome libraries were sequenced on an Illumina HiSeq platform (Illumina, Inc., San Diego, CA, USA) with 75 bp paired-end reads at the Northwest Genomics Center at the University of Washington (Seattle, WA, USA). Sequence reads were aligned to the hg19 reference genome using Burrows-Wheeler Aligner (BWA) version 0.7.8. Variants were called by the GATK unified genotyper, version 3.1.1, and annotated with SeattleSeq, version 138. Variants not predicted to impact protein-coding sequence or present (AC > 3) in the Genome Aggregation Database, gnomAD version 2.0, were excluded. All putative de novo variants were validated using Sanger sequencing.

The LGD mutations in patients CC1.p1, TI.p1, AA.p1, and GU.mo were detected using smMIPs. The smMIP sequencing and analysis was performed as previously described[19,20] and summarized above. For patient CC1.p1, the splice-site mutation was detected using smMIP-based targeted sequencing of 2514 probands with a primary diagnosis of ASD. For patient TI.p1, the disruptive mutation was detected using smMIP-based targeted sequencing of 1201 probands with primary diagnosis of DD. For patient AA.p1, the disruptive mutation was detected using smMIP-based targeted sequencing of 2383 probands with primary diagnosis of ID. For patient GU.mo, the disruptive mutation was detected using smMIP-based targeted sequencing of 253 probands with primary diagnosis of ID. Validation and inheritance determination were performed by Sanger sequencing in both probands and parents.

aCGH was applied for detection of variants in individuals DU.p1, PI.p1, CF.p1, and CF.p2. The de novo CNV in patient DU.p1 was detected using customized exon-targeted oligo array (OLIGO V8.1), which covers more than 1700 disease-associated genes, respectively, with exon-level resolution[63] at Baylor Genetics. Chromosomal microarray analysis (CMA) revealed a copy number loss of chromosome band 17q23.3 of approximately 0.146 Mb in size (17:61271293–61417468, GRCh37) followed by a copy number gain of chromosome band 17q25.1 of approximately 0.316 Mb in size (17:72205984–72521915) in the proband. The clinical significance is unclear at the present time. Parental CMA revealed the copy number loss of chromosome 17q23.3 is not present in either parent. Parental CMA also showed that the copy number gain of chromosome 17q25.1 is present in the mother. This copy number gain most likely represents a familial variant although the clinical significance of this gain is unclear at the present time.

The de novo CNV in patient PI.p1 was detected using aCGH 4 × 44K kit (Agilent, Santa Clara, CA, USA) at the Laboratories of Clinical Genetics and Molecular Genetics of Neurodevelopment, Department of Women's and Children's Health, University of Padua, Italy. aCGH was performed according to standard protocols. Further analysis with CGH 244K kit and real-time PCR was performed on the proband's sample to refine the breakpoints of the detected alteration. Array results were analyzed with DNA Analytic 4.0 Software (Agilent, Santa Clara, CA, USA) and genomic positions were based on Human Genome Assembly NCBI Build 37/hg19.

The CNV in patients CF.p1 and CF.p2 was detected using an aCGH 8 × 60k kit (Agilent, Santa Clara, CA, USA) at CHU Lille, Clinique de Génétique Guy Fontaine, Lille, France. aCGH was performed according to standard protocols. Further analysis with a FISH study with BAC RP11-191F21 was performed to detect the mosaic status[64] of the father CF.fa (9 of the 150 mitoses).

**Analysis of TANC2 expression in the developing human brain**. To interrogate the expression pattern of TANC2 and TANC1 in the developing human cerebral cortex, we analyzed a single-cell RNA sequencing dataset generated from across 48 individuals provided in a recent publication[26]. In short, processed normalized expression values, and the associated metadata, were downloaded from the UCSC cell browser (https://cells.ucsc.edu/dev/?ds=cortex-dev). Cluster assignments for each cell were directly used from the source study, and biological interpretation of each broad cell type was made based on the published analysis and is further summarized in Supplementary Table 3. For each cell type, we calculated enrichment of TANC2 or TANC1 using an odds ratio, and statistical significance of enrichment or depletion was calculated using Wilcoxon rank sum test (using wilcox.test() from R stats v3.5.3), followed by multiple hypothesis correction using Bonferroni adjustment (p.adjust() built in function in R).

**Drosophila stocks**. D. melanogaster stocks were cultured in standard medium at 25 °C. The following stocks were obtained from the Bloomington Drosophila Stock Center (BDSC): UAS-rols-RNAi lines—$y^1sc^*v^1$; P{TRiP.HMC04426}attP40 (BDSC_56986) and $y^1sc^*v^1$; P{TRiP.HMJ22326} attP40(BDSC_58262), UAS-rols - $y^1$ $w^*$; P{Mae-UAS.6.11}rolsLA00796 (BDSC_22194), rols deficiency line - $w^{1118}$; Df (3 L)ED4475, P{3'.RS5+3.3'}ED4475/TM6C, $cu^1$ $Sb^1$ (BDSC_8069), UAS-mCD8:: GFP - $w^*$; P{10XUAS-mCD8::GFP}attP2 (BDSC_32184), UAS-nls::GFP - $w^{1118}$; P {UAS-GFP.nls}8 (BDSC_4776), G-TRACE - $w^*$; P{UAS-RedStinger}4, P{UAS-FLP. D}JD1, P{Ubi-p63E(FRT.STOP)Stinger}9F6/CyO (BDSC_28280), Repo-Gal4 - $w^{1118}$; P{GAL4}repo/TM3, $Sb^1$ (BDSC_7415), C57-Gal4 was previously described[64]. The newly generated $y^1$ $w^*$; Mi{Trojan-GAL4.1}rols$^{MI02479-TG4.1}$/TM3, $Sb^1$ $Ser^1$ (BDSC_76150), $y^1$ $w^*$; Mi{PT-GFSTF.1}rols$^{MI02479-GFSTF.1}$ (BDSC_64471), and UAS-TANC2 - $y^1$ $w^*$; PBac{UAS-hTANC2.B}VK37 (BDSC_78452) were generated in the Bellen lab and have been deposited in BDSC. The 80 kb genomic rescue line for rols was generated by Genetivision by insertion of the CH321-52G11 P[acman] genomic BAC clone into the VK37 landing site on chromosome 2. Study protocols comply with all relevant ethical regulations and were approved by the IRB of Central South University.

**Imaging and NMJ analysis**. Whole-mount immunostaining of the Drosophila NMJ was performed[31,65]. For adult brains: adult flies were dissected and fixed in 4% PFA in PBS overnight at 4 °C. Brains were transferred to 2% Triton X-100 in PBS (phosphate-buffered saline) at room temperature for permeabilization and subsequently vacuumed for 1 hr then incubated overnight at 4 °C. Third instar larval brains were fixed in 4% PFA in PBS at 4 °C for 2 h and transferred to 0.5% Triton X-100 in PBS at 4 °C overnight. All tissues were blocked in 10% normal goat serum with 0.5% Triton X-100 in PBS and incubated with primary antibodies: anti-GFP conjugated with FITC ab6662 (Abcam), 1:500; anti-Elav (Embryonic lethal abnormal vision) rat monoclonal: 7E8A10 (DSHB), 1:200; anti-Repo (mouse monoclonal: 8D12 (DSHB), 1:50. Primary antibodies were incubated at 4 °C overnight and then washed 3× with 0.5% Triton X-100 in PBS. For Elav staining in larval brains, the secondary antibody conjugated to Alexa-647 number: 712-605-153 (Jackson ImmunoResearch) and CY3 number: 712-165-153 (Jackson ImmunoResearch) conjugated was used for the adult brains. The secondary for Repo staining was Alexa-647 number: 715-165-150 (Jackson ImmunoResearch). The secondary for HRP used was anti-rabbit Alexa-405 number: 711-475-152 (Jackson ImmunoResearch). All secondaries were diluted 1:250 in 0.5% Triton X-100 in PBS and incubated at 4 °C overnight, washed 4× and mounted in RapiClear® (#RC147001—SunJin Lab Co.), and imaged with confocal microscopy (Zeiss LSM880). Images were processed using Imaris.

For NMJ analysis, primary and secondary antibodies were used: rabbit anti-HRP (1:1000; code number: 323-005-021; Jackson ImmunoResearch, West Grove, PA, USA); mouse anti-DLG (4F3; 1:50; DSHB); mouse anti-GluRIIA (8B4D2; 1:50; DSHB); Alexa 488- or cy3-conjugated anti-mouse and anti-rabbit secondary antibodies (1:500; Jackson ImmunoResearch, West Grove, PA, USA). All images were collected using a ZEISS LSM880 confocal microscope and analyzed with image J software (National Institutes of Health). Statistical analysis was performed using GraphPad Prism 6.0 software (GraphPad Software, Inc.). Statistical significance was performed using one-way ANOVA for comparisons of all group means. Data expressed as the means ± SEM $p$ values <0.05 were considered to be statistically significant. The analysis was performed double blind.

**Behavior testing**. Housing and handling—All flies were grown in a temperature and humidity controlled incubator at 25 °C and 50% humidity on a 12-h light/dark cycle. Flies were reared on standard fly food (water, yeast, soy flour, cornmeal, agar, corn syrup, and propionic acid). Collection of socially naïve adults was performed by isolating pupae in 16 × 100 polysterine vials containing approximately 1 mL of fly food. After eclosion, flies were anesthetized briefly with $CO_2$ to ensure they were healthy and lacked wing damage. Anesthetized flies were returned to their vials and allowed a full 24 h to recover before testing.

Courtship paradigm—Courtship assays were performed in a custom-made six-well plate with 40 mm circular wells, with a depth of 3 mm and 11° sloped walls (milled[66]). One Canton-S virgin female (aged 6–10 days) and experimental male (aged 3–5 days) were simultaneously introduced into the chamber via aspiration. Recordings were taken using a Basler 1920UM, 1.9MP, 165FPS, USB3 Monochromatic camera using the BASLER Pylon module, with an adjusted capturer rate of 33FPS. Conversion of captured images into a movie file was performed via a custom MatLab script, and tracking of flies in the movie was

performed using the Caltech Flytracker[37]. Machine learning assessment of courtship was performed using JAABA[36] and 20% videos were manually verified for accuracy post screening.

Software and statistics—Data analysis was performed using Microsoft Excel and GraphPad Prism. Determination of significance in behavior tests was performed using the Kruskal–Wallis one-way analysis of variance and the Dunn's multiple comparison test. $P$ values of 0.05 or less were considered significant. Outliers were identified and eliminated using a ROUT test ($Q = 1\%$). Evaluation of significance when examining the number of successful copulations was determined via Chi-Square test. $P$ values of 0.05 or less were considered significant.

**Climbing and bang-sensitivity assays.** Flies were anesthetized with $CO_2$ and housed in individual vials 24 h prior to testing[67]. For climbing, flies were given 1 min to habituate to an empty vial and tapped to induce a climbing response and timed to reach the 7 cm mark. For bang-sensitivity, flies were vortexed for 15 s and assessed for recovery time to an upright position.

**URLs.** For GeneMatcher, see https://www.genematcher.org/; for ANNOVAR, see http://annovar.openbioinformatics.org/en/latest/; for MARRVEL, see https://www.marrvel.org.

**Reporting Summary.** Further information on research design is available in the Nature Research Reporting Summary linked to this article.

## Data availability
The MIP sequencing data for this study have been deposited in the NIMH data repository National Database for Autism Research (NDAR) (https://doi.org/10.15154/1252218) and is available to all qualified researchers after data use certification. A reporting summary for this article is available as a Supplementary Information file. Underlying data in Figs. 1a; 4; 6b, c; 7d–f; Supplementary Figs. 5, 6a, b are provided as Source Data file.

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

## Acknowledgements

We thank Tonia Brown for assistance in editing this manuscript. We thank all of the families participating in this study. We are grateful to all of the families at the participating SSC sites, as well as the principal investigators (A. Beaudet, R. Bernier, J. Constantino, E. Cook, E. Fombonne, D. Geschwind, R. Goin-Kochel, E. Hanson, D. Grice, A. Klin, D. Ledbetter, C. Lord, C. Martin, D. Martin, R. Maxim, J. Miles, O. Ousley, K. Pelphrey, B. Peterson, J. Piggot, C. Saulnier, M. State, W. Stone, J. Sutcliffe, C. Walsh, Z. Warren, E. Wijsman). We appreciate obtaining access to phenotypic data on SFARI Base. Approved researchers can obtain the SSC population dataset described in this study (https://www.sfari.org/resource/resources/simons-simplex-collection/) by applying at https://base.sfari.org. We would like to thank Drs. Emanuela Leonardi, Roberta Polli, Chiara Rigon, Ornella Galesi, and Giuseppe Calabrese for useful discussions and technical assistance. This work was supported by the following grants: the National Natural Science Foundation of China (NSFC 31671114, 81871079) to H.G.; Simons Foundation Autism Research Initiative (SFARI 303241) to E.E.E.; National Institutes of Health (NIH R01MH101221) to E.E.E.; NIH (R01MH100047) to R.A.B.; NIH (R01NS069605) to H.C.M.; NSFC (81330027, 81525007) to K.X.; Key R&D Program of Hunan Province (2018DK2016) to Z.H.; SFARI (491371) to T.J.N.; and a grant from the South Carolina Department of Disabilities to C.E.S. E.E.E. and H.J.B. are investigators of the Howard Hughes Medical Institute. This work is also supported by NIH (U54NS093793) to H.J.B., S.Y. and M.J.W.; H.J.B. is supported by NIH (R01GM067858, R24OD022005); S.Y. and M.F.W. by a Simons Foundation Functional Screen Award (368479). M.A.G. was supported by the U.S. National Institutes of Health (T32HG000035). We would like to thank Hongling Pan, Danqing Bei and Kai Yuan's lab for technical assistance. Confocal microscopy at Baylor College of Medicine is supported in part by NIH grant U54HD083092 to the Intellectual and Developmental Disabilities Research Center (IDDRC) Neurovisualization Core. P.C.M. is funded by CIHR and the Stand by Eli Foundation. J.C.A. is supported by NIH F32 (NS110174-01). WES for proband MA.p1 was provided by the University of Washington Center for Mendelian Genomics (UW-CMG) and funded by the National Human Genome Research Institute and the National Heart, Lung and Blood Institute grant HG006493 to Drs. Debbie Nickerson, Michael Bamshad, and Suzanne Leal. The content is solely the responsibility of the authors and does not necessarily represent the official views of the NIH.

## Author contributions

H.G., E.B., P.C.M., A.P.A.S., A.M., and E.E.E. conceived the study. H.G., E.B., P.C.M., J.C.A., H.J.B., A.M. and E.E.E. wrote and revised the manuscript. All the authors read and commented on the manuscript. H.G. and K.X. analyzed the SSC and ASC WES data and designed smMIP-targeted genes and supervised the smMIP analysis. P.C.M., J.A., H.J.B., S.Y. and M.J.W. generated transgenic flies, conducted imaging (with S.J.), performed behavioral assays and edited the manuscript. R.Z. performed the *Drosophila* NMJ analysis. T.J.N. performed the single-cell RNA-seq data analysis. T.W., H.W. and M.L. performed the smMIP experiments and variant calling; H.W., M.L. and W.Z. validated smMIP data analysis. Other authors participated in generating the phenotypic and genetic data for the patients and providing expert clinical interpretation and details of the phenotype for each affected individual.

## Competing interests

E.E.E. is on the scientific advisory board (SAB) of DNAnexus, Inc. The Department of Molecular and Human Genetics at Baylor College of Medicine receives revenue from clinical genetic testing, including exome sequencing and microarray analysis, at Baylor Genetics. J.J. is an employee of GeneDx, Inc. The remaining authors declare no competing interests.

## Additional information

Hui Guo [1,2,38], Elisa Bettella[3,4,38], Paul C. Marcogliese [5,6,38], Rongjuan Zhao[2], Jonathan C. Andrews[5,6], Tomasz J. Nowakowski [7,8,9], Madelyn A. Gillentine[1], Kendra Hoekzema[1], Tianyun Wang[1,2], Huidan Wu[2], Sharayu Jangam[5,6], Cenying Liu[2], Hailun Ni[2], Marjolein H. Willemsen[10,11], Bregje W. van Bon[10], Tuula Rinne[10], Servi J.C. Stevens[11], Tjitske Kleefstra[10], Han G. Brunner[10,11], Helger G. Yntema[10], Min Long[2], Wenjing Zhao[2], Zhengmao Hu[2], Cindy Colson[12], Nicolas Richard [12], Charles E. Schwartz [13], Corrado Romano [14], Lucia Castiglia[14], Maria Bottitta[14], Shweta U. Dhar[5], Deanna J. Erwin[5], Lisa Emrick[5], Boris Keren[15], Alexandra Afenjar[16], Baosheng Zhu[17,18], Bing Bai[17,18], Pawel Stankiewicz[5], Kristin Herman[19], University of Washington Center for Mendelian Genomics, Saadet Mercimek-Andrews [21], Jane Juusola[22], Amy B. Wilfert [1], Rami Abou Jamra [23], Benjamin Büttner[23], Heather C. Mefford[20], Alison M. Muir [20], Ingrid E. Scheffer[24], Brigid M. Regan[24], Stephen Malone[25], Jozef Gecz [26], Jan Cobben[27,28], Marjan M. Weiss [29], Quinten Waisfisz[29], Emilia K. Bijlsma[30], Mariëtte J.V. Hoffer[30], Claudia A.L. Ruivenkamp[30], Stefano Sartori [31], Fan Xia [5], Jill A. Rosenfeld[5], Raphael A. Bernier[32], Michael F. Wangler[5,6,33], Shinya Yamamoto [5,6,33,34], Kun Xia[2,35], Alexander P.A. Stegmann[10,11], Hugo J. Bellen [5,6,33,34,36], Alessandra Murgia [3]* & Evan E. Eichler [1,37]*

[1]Department of Genome Sciences, University of Washington School of Medicine, Seattle, WA 98195, USA. [2]Center for Medical Genetics & Hunan Key Laboratory of Medical Genetics, School of Life Sciences, Central South University, 410078 Changsha, Hunan, China. [3]Laboratory of Molecular Genetics of Neurodevelopment, Department of Women's and Children's Health, University of Padua, Via Giustiniani 3, 35128 Padua, Italy. [4]Fondazione Istituto di Ricerca Pediatrica Città della Speranza, Corso Stati Uniti 4, 35129 Padua, Italy. [5]Department of Molecular and Human Genetics, Baylor College of Medicine, Houston, TX 77030, USA. [6]Jan and Dan Duncan Neurological Research Institute, Texas Children's Hospital, Houston, TX 77030, USA. [7]UCSF Department of Anatomy, University of California, San Francisco, San Francisco, CA 94143, USA. [8]UCSF Department of Psychiatry, University of California, San Francisco, San Francisco, CA 94143, USA. [9]UCSF Weill Institute for Neurosciences, University of California, San Francisco, San Francisco, CA 94158, USA. [10]Department of Human Genetics, Radboud University Medical Center, 6500 HB Nijmegen, The Netherlands. [11]Department of Clinical Genetics, Maastricht University Medical Center, 6202 AZ Maastricht, The Netherlands. [12]Normandie Univ, UNICAEN, CHU de Caen Normandie, Department of Genetics, EA7450 BioTARGen, 14000 Caen, France. [13]Greenwood Genetic Center, Greenwood, SC 29646, USA. [14]Oasi Research Institute-IRCCS, 94108 Troina, Italy. [15]Département de génétique, Hôpital Pitié-Salpêtrière, Assistance Publique - Hôpitaux de Paris, 75013 Paris, France. [16]APHP, Centre de référence des malformations et maladies congénitales du cervelet Département de génétique et embryologie médicale, GRCn°19, pathologies Congénitales du Cervelet-LeucoDystrophies, AP-HP, Hôpital Armand Trousseau, F-75012 Paris, France. [17]Department of Pediatrics, The First People's Hospital of Yunnan Province, 650032 Kunming, Yunnan, China. [18]Medical Faculty, Kunming University of Science and Technology, 650032 Kunming, Yunnan, China. [19]Section of Medical Genetics, Medical Investigation of Neurodevelopmental Disorders Institute, University of California, Davis, Sacramento, CA 95817, USA. [20]Department of Pediatrics, Division of Genetic Medicine, University of Washington, Seattle, WA 98195, USA. [21]Division of Clinical and Metabolic Genetics, Department of Pediatrics, University of Toronto, The Hospital for Sick Children, Toronto, ON M5G 1X8, Canada. [22]GeneDx, Gaithersburg, MD 20877, USA. [23]Institute of Human Genetics, University of Leipzig Medical Center, Leipzig, Germany. [24]Departments of Medicine and Paediatrics, The University of Melbourne, Austin Health and Royal Children's Hospital, Melbourne, VIC 3084, Australia. [25]Department of Neurosciences, Queensland Children's Hospital, Brisbane, QLD 4101, Australia. [26]School of Medicine and the Robinson Research Institute, The University of Adelaide at the Women's and Children's Hospital, Adelaide, SA 5006, Australia. [27]Emma Children's Hospital AUMC, 1105 AZ Amsterdam, The Netherlands. [28]North West Thames Genetics Service NHS, London, UK. [29]Amsterdam UMC, Vrije Universiteit Amsterdam, Department of Clinical Genetics, Amsterdam, Netherlands. [30]Department of Clinical Genetics, Leiden University Medical Center, 2333 ZA Leiden, The Netherlands. [31]Paediatric Neurology and Neurophysiology Unit, Department of Women's and Children's Health, University Hospital of Padua, 35128 Padua, Italy. [32]Department of Psychiatry, University of Washington, Seattle, WA 98195, USA. [33]Program in Developmental Biology, Baylor College of Medicine, Houston, TX 77030, USA. [34]Department of Neuroscience, Baylor College of Medicine, Houston, TX 77030, USA. [35]Hunan Key Laboratory of Animal Models for Human Diseases, 410078 Changsha, Hunan, China. [36]Howard Hughes Medical Institute, Baylor College of Medicine, Houston, TX 77030, USA. [37]Howard Hughes Medical Institute, University of Washington, Seattle, WA 98195, USA. [38]These authors contributed equally: Hui Guo, Elisa Bettella, Paul C. Marcogliese. A full list of consortium members appears at the end of the paper.

## University of Washington Center for Mendelian Genomics

Deborah A. Nickerson[1] & Michael J. Bamshad[20]

