## [Peer Review File · Nature Communications]

Reviewers' Comments:

Reviewer #1:

Remarks to the Author:

Remarks to the authors

In this manuscript, Guo, Bettella and colleagues perform a meta-analysis of de novo mutations identified through WES, WGS, or targeted gene sequencing across published ASD studies and center on a single gene, TANC2, that they decide to follow-up the clinical outcomes. While the clinical follow-up is deep, the initial underlying statistical evidence behind TANC2 isn't strong enough to warrant the follow-up (and falls prey to the same flawed statistical practices in Stessman et al., Nature Genetics 2017). Ultimately, the lack of statistical rigor begets asking why the gene was followed-up in the first place. Based on these concerns, I do not feel this work merits publication in Nature Communications and seems more appropriate for a clinical genetics focused journal.

Major comments

1) The authors noted that they combined the published de novo mutations in the SSC with those from two other cohorts, ASC and MSSNG, to improve power for discovery. While in general increasing sample size will increase power, it is important to ensure proper quality control and consistency across the different studies. Without demonstrating the underlying data is sound and free of technical artifacts (e.g., different sequencing depth between studies, different capture kits, different sequencing technology, different QC measures for calling de novo mutations), the subsequent results may be compromised. As such, I strongly suggest doing the following analyses to ensure the validity of all subsequent findings:

- Evaluation of between-study heterogeneity (e.g. overall mutation rate across studies). Does each cohort (SSC, ASC, and MSSNG) contribute a similar overall number of coding de novo mutations per individual? A pair-wise Poisson exact test (`poisson.test` in R) will suffice. Cohorts with significantly more or fewer de novo mutations should be removed from the analysis and may improve subsequent findings.

- Analysis of negative control mutation type: previous studies often include synonymous mutations under the hypothesis that, as a class, synonymous mutations do not confer risk to disease (i.e., equal frequencies among cases and controls). Again, it would be reassuring to see a similar frequency of de novo synonymous mutations both between cohorts and between probands and siblings. If there is a significant difference, it would be better to either remove offending studies or normalize the data by the difference in the frequency of de novo synonymous mutations.

- It is worth noting, the MSSNG WGS cohort described in Yuen et al., Nature Neuroscience 2017 was sequenced on two different platforms, Illumina and Complete Genomics. The number of SNVs and indels appeared to differ significantly between technologies (Figure 2A and 2B of the manuscript) and thus the inclusion of this cohort may lower power for association due to the heterogeneous nature of the underlying data (as fewer variants were detected on the Complete Genomics platform).

2) I must be missing something as I struggled to understand exactly how the gene prioritization methodology worked that culminated in 71 genes in Table 1. It came across that the only conditions were 1) that the gene was a target of RMRP or RBOX and 2) pass the pLI and RVIS cutoffs but then it wasn't clear how the de novo mutations (from SSC, ASC, and MSSNG) were involved.

3) Combining published de novo mutations from WES and WGS with smMIP-sequencing but only correcting for 14 genes is statistically invalid, as Barrett et al., bioRxiv 2016 pointed out (<https://www.biorxiv.org/content/early/2017/07/07/115964>) when the same analysis strategy was done in Stessman et al., Nature Genetics 2016 (doi: 10.1038/ng.3792). As Barrett et al., pointed out, one can either: a) combine the data but correct for all genes in the genome or b) use

only the smMIP-seq de novos and correct for those genes in the second wave (i.e., 14 genes). Lastly, because the authors used two different mutational models for each of these 14 genes, they need to correct for each gene twice, not once.

3) Could the authors explain the rationale for why those 14 specific genes were selected for smMIP-sequencing as opposed to any of the other 57 genes.

4) It was unclear why the authors tested two different mutational models given that this imposes an additional testing correction and doesn't appear to provide any additional insight.

It would be appreciated if the authors would add the following citations:

- pLI (Lek et al., Nature 2016) (line 117)
- RVIS (Petrovski et al., PLOS Genetics 2013) (line 117)
- ExAC was used by not cited (line 103) (Lek et al., Nature 2016)
- Filtering De novo mutations using ExAC should be cited (Kosmicki et al., Nature Genetics 2017)
- 1000 Genomes Project (line 160)
- ESP6500 (line 160)

Minor comments:

- On line 180 and 181, I assume instead of another five case had no autistic phenotypes the authors meant the remaining five cases had no autistic phenotypes? Also, it should be cases not case.
- Could the authors add confidence intervals to Figures 1a and 1b?
- Which cohorts comprise the 8 cohorts specified on lines 164-165?

Reviewer #2:

Remarks to the Author:

In this manuscript, Guo et al. undertake a large scale, systematic study to identify and characterize human mutations in the gene TANC2 in neurodevelopmental disorders. The sheer amount of experimental and logistical work involved in this human disease project is impressive. In particular, they provide both genetic and clinical data for 20 individuals identified with mutations in TANC2 and provide evidence that these mutations are associated with a spectrum of neurological and neuropsychiatric diseases, including intellectual disability, autism spectrum disorder, epilepsy, and others. Finally, because TANC2 interacts with components at the postsynaptic density, they go on to study *Drosophila* neuromuscular junctions. Using neuronal RNAi knockdown of the fly homolog of TANC2, *rols*, the authors report a moderate increase in synaptic growth. Together, they conclude disruption of TANC2 contributes to a variety of seemingly disparate neurodevelopmental diseases.

Although TANC2 has been implicated in neurodevelopmental diseases in humans previously, the identification of new human disease-causing alleles of TANC2 and their clinical description are important. The PSD is well known to be a key target for many neurodevelopmental, neurological, and psychiatric diseases, and the identification of specific alleles of TANC2 and their clinical manifestations is exciting. However, the characterization of *rols* in *Drosophila* is very preliminary and needs major additional work. In particular, several key genetic experiments, interpretations, and additional assays need to be performed to properly assess the role of *rols* at synapses, as well as to validate and extend their insights. Without these experiments, there is little additional insight into how TANC2/*Rols* may impact synaptic development and/or function.

Major points:

1. One major question the authors fail to consider is that TANC2/*Rols* is a PSD interacting protein with clear roles in dendrites (the authors cite the relevant papers here and underscore this point in

the abstract). However, at the fly NMJ, the muscle is the postsynaptic compartment with high expression of PSD95 (DLG) and other postsynaptic components. However, the authors reduce *rols* expression in presynaptic neurons (both pan-neuronal and motor-neuron RNAi knockdown). How do the authors interpret a presynaptic function of *rols* in regulating synaptic growth (through an undermined mechanism) with the established functions of TANC2 at the PSD? This seems to be a major conundrum that is not discussed or considered.

2. There are several important controls in assessing synaptic growth at the NMJ and the function of *rols* that should be performed. Fortunately, these experiments should be relatively straightforward. First, there are 2 *rols* RNAi lines available from the TRiP library at the Bloomington Drosophila Stock Center. It is not clear from the methods whether the single RNAi line they used is one of these, but regardless the second RNAi lines as well as additional neuronal drivers (*nsyb-Gal4*, *D42-Gal4*, etc) can be used. In addition, because TANC2 is a PSD-interacting protein, and many studies on fly *rols* have demonstrated functions in the postsynaptic muscle, postsynaptic RNAi of *rols* (using *G14-Gal4*, *MHC-Gal4*, etc) should be performed and synaptic growth assessed. More importantly, a variety of loss-of-function *rols* alleles (*ex18*, etc) and UAS-*rols* transgenic lines have been generated and are available. These genetic reagents can be used to assess NMJs of *rols* mutants and to rescue any phenotypes observed by expression of *rols* in neurons or muscle.

3. There are a variety of additional experiments that could be performed to gain insight into *rols* function at the NMJ. Ideally, electrophysiological recordings would be performed in *rols* mutants/RNAi lines to assess impacts on synaptic function, although this Reviewer understands this approach may be beyond the scope and toolkit of the authors. However, there appears to be reagents available from public stock centers as well as from other investigators to assess *rols* expression (*rols* promoter-Gal4 fusions), transgenic overexpression/rescue lines (UAS-*rols* as well as UAS-TANC2), as well as perhaps tagged constructs and/or antibodies. These could be used to reveal the subcellular localization of *Rols* at the NMJ (at the PSD in muscle and/or in motor neurons) and to assess whether overexpression of *rols* impacts synaptic growth (perhaps inhibiting growth?). These experiments could provide more insight into how disruption of *rols* impacts synaptic growth or function. At minimum, additional immunostaining against presynaptic active zones and postsynaptic glutamate receptors using established antibodies can be performed to better detail NMJ synapses in *rols* mutants/RNAi lines. Since *rols* is a putative PSD gene, was the organization of DLG altered at the PSD in *rols* RNAi? Or mutants?

Minor point:

1. Typo on line 196

Reviewer #3:

Remarks to the Author:

This paper describes what was initially a refreshingly simple approach to plausible developmental disease gene identification. This begins with the Simons Simplex quad whole exome data. The authors have compared the de novo likely gene disruptive events on a per gene and combined basis so confirm the excess of LGD events in the probands remains when the known genes are excluded. The then filtered the genes by their reported association as mRNA with FMRP etc to show further enrichment. The candidate genes were selected as high pLi and RVIS levels chosen on the basis of a 1 in 20 false discovery rate. This is all sensible and standard approach. This approach became significantly less easy to assess and it was then repeated using a combined cohort including two groups with which I am unfamiliar, particularly with the case mix or repertoire of known causative gene de novo mutations in ASC and MSSNG. This led to 71 genes being targeted for further analysis (many of which would be considered known neurodevelopment disorder genes).

The paper then shifts to concentrate on TANC2 in which only one LGD DNM had been identified in SSC to sequencing of a further cohort and to expand the analysis to include DNM that are NSV.

Matchmaker was used to assemble a total of 19 families which includes 3 intragenic CNV. 11 of the LGD events were de novo, one inherited from a music parent. In the other inherited cases are taken as evidence for a strong psychiatric predisposition associated with carriers of LGD events

This is an interesting paper and TANC2 may indeed represent a genuinely causative NDD gene. However I cannot judge on the basis of the presented data how likely this is to be due to ascertainment bias alone. The phenotypic information is not compelling as there is no indication of how unusual the "pattern" is compared to randomly selected groups of 20 cases from the myriad of cohorts represented in the mutation table.

I accept that the lack of LGD events in ExAC and gnomAD is interesting but there are 8 LGD events as this is a highly biased sample which has been selected for the presence of a LGD DNM in TANC2. The main deficiency is the lack of a very large scale cohort that is genuinely unselected for disease.

It would be useful to have some idea of the total number of cases screened to generate these 19 families - and the distribution of clinical features. However I accept that this is an extremely difficult task so my preference would be for a more circumspect approach to the assignment of pathogenesis.

Reviewers' comments:

Reviewer #1 (Remarks to the Author):

Remarks to the authors

In this manuscript, Guo, Bettella and colleagues perform a meta-analysis of *de novo* mutations identified through WES, WGS, or targeted gene sequencing across published ASD studies and center on a single gene, *TANC2*, that they decide to follow-up the clinical outcomes. While the clinical follow-up is deep, the initial underlying statistical evidence behind *TANC2* isn't strong enough to warrant the follow-up (and falls prey to the same flawed statistical practices in Stessman et al., Nature Genetics 2017). Ultimately, the lack of statistical rigor begets asking why the gene was followed-up in the first place. Based on these concerns, I do not feel this work merits publication in Nature Communications and seems more appropriate for a clinical genetics focused journal.

Two lines of evidence support mutation of *TANC2* and neurodevelopmental disease. First, we observe a significant enrichment of *de novo* mutations in *TANC2* even after adjusting for the total number of genes in the genome. We identified 9 (now revised to 10) *de novo* truncating mutations from 16,113 autism and developmental delay patients. Assessing only for the frequency of *de novo* truncating mutations, we find a significant excess of *de novo* LGD mutations whether we use the chimpanzee-human model for divergence or the denovolyzeR model, which considers mutational context. This analysis survives multiple-text correction considering 18,946 genes in chimpanzee-human model and 19,618 in denovolyzeR model ($P_{\text{adj}} = 4.6 \times 10^{-10}$, CH model; $P_{\text{adj}} = 9.4 \times 10^{-10}$, denovolyzeR model, Bonferroni correction). This finding is also robust to differences in exome sequence coverage and differences in *de novo* mutation rate sensitivity.

As a second test, we also performed a more traditional burden analysis between cases and controls (Fisher's exact test). For example, we identified 13 probands carrying LGD mutations with a known cohort size ($n = 17,567$), which represents a nominally significant enrichment of LGD mutations in NDD patients when compared to 45,375 ExAC nonpsychiatric control samples. To guard against potential exon dropout, we assessed the mean coverage of *TANC2* exons in ExAC. The average coverage is 49.2 sequence reads per exon (Supplementary Fig. 3) with 24/25 of the exons showing on average more than 20-fold sequence read coverage.

It is true that initial analysis of the SSC and ACGC identified only two LGD mutations but given the gene's involvement in the postsynapse and the extremely rare LGD frequency in controls, we targeted a larger set of patients both by resequencing and by exome analysis. This strategy is similar to our approach

with genes *CHD8*, *ADNP*, *DYRK1A*, *POGZ*, etc. where we only observed a handful of cases but then followed up in much larger cohorts establishing these as now well-known autism risk genes.

We did not repeat the same error that was performed in an earlier analysis (Stessman et al., 2017) but stated that correcting for the total number of genes in the genome would result in an insignificant result. We believe some of the confusion regarding our statistical analysis stems from the sentence below, which we have now revised:

“Combining the *de novo* mutations in *TANC2* identified in the SSC cohort (1 LGD and 1 missense), we found an excess of *de novo* LGD mutations correcting for 14 target genes ($P = 0.028$, CH model; $P = 0.032$, DenovolyzeR model) (Method), although significance would not survive genome-wide multiple-test correction.”

Revised text:

“Combining the *de novo* mutations in *TANC2* identified in the SSC cohort (1 LGD and 1 missense), we observed a trend for an excess of LGD mutations in probands although this observation did not remain significant after genome-wide multiple-test correction.”

Major comments

1) The authors noted that they combined the published *de novo* mutations in the SSC with those from two other cohorts, ASC and MSSNG, to improve power for discovery. While in general increasing sample size will increase power, it is important to ensure proper quality control and consistency across the different studies. Without demonstrating the underlying data is sound and free of technical artifacts (e.g., different sequencing depth between studies, different capture kits, different sequencing technology, different QC measures for calling *de novo* mutations), the subsequent results may be compromised. As such, I strongly suggest doing the following analyses to ensure the validity of all subsequent findings:

The reviewer raises a fair point. We have gathered available data from these three initial studies and compared *de novo* mutation rate estimates, which are generally comparable. The SSC serves as a general benchmark for comparison because the same individuals have been analyzed independently using whole-genome sequence data, which reveals an undercalling of ~2.5% in the SSC exomes (Turner et al., Cell, 2017).

Cohorts	Phenotype	#Samples	LGD DNM	Missense DNM	Synonymous DNM	Overall DNM Rate	Synonymous DNM rate
SSC (Iossifov2014)	ASD	2,508	384	1,673	643	1.08	0.256
SSC (Iossifov2014)	Sibling	1,902	175	1,128	483	0.94	0.254
ASC (DeRubeis2014)	ASD	1,445	192	1,107	390	1.17	0.270
MSSNG (Yuen2017)	ASD	1,625	232	1,188	484	1.17	0.298

We cannot access the underlying exome sequence data underlying either ASC or genome sequence data from MSSNG. We have specifically requested that for the ASC but apparently consent issues preclude such data from being distributed. Wherever possible, however, we have tried to control for potential confounders in this analysis. For example, we excluded all MSSNG WGS samples that were generated by two different sequencing platforms, i.e., Illumina vs. Complete Genomics. We considered genes where there was significant representation in both genomes and exome capture kits. In comparisons with ExAC, we restricted the analysis to exons where there was minimum of 20-fold sequence coverage. We believe our treatment of the data are conservative since the general observation has been undercalling of genetic variants as opposed to overcalling of DNMs. **In our study we used these three cohorts (SSC, ASC, MSSNG; now reduced to two: SSC and ASC) simply for the purpose of filtering candidate genes based on FMRP/RBFOX binding or intolerance to mutation (pLI/RVIS). Undercalling would generally lead to false negative as opposed to false positive signals.**

- Evaluation of between-study heterogeneity (e.g. overall mutation rate across studies). Does each cohort (SSC, ASC, and MSSNG) contribute a similar overall number of coding de novo mutations per individual? A pair-wise Poisson exact test (`poisson.test` in R) will suffice. Cohorts with significantly more or fewer *de novo* mutations should be removed from the analysis and may improve subsequent findings.

We compared the exonic DNM rates between studies and although they are very similar they are not identical (SSC vs. ASC: $p = 0.003$, SSC vs. MSSNG: $p < 2.2 \times 10^{-16}$, Poisson test). The lowest mutation was observed for the SSC where, as mentioned, comparisons with whole-genome datasets suggest we are undercalling (2.5%) DNMs. We also considered other DNM data from two large-scale WES studies of developmental delay (DDD, Nature, 2017) and intellectual disability (Lelieveld et al., Nat Neurosci, 2016). This comparison establishes a rate for exonic DNM ranging from 1.08-1.80 (see table below).

Studies	Phenotype	#Samples	LGD DNM	Missense DNM	Synonymous DNM	Overall DNM Rate	Synonymous DNM rate
SSC (Iossifov2014)	ASD	2,508	384	1,673	643	1.08	0.256
ASC (DeRubeis2014)	ASD	1,445	192	1,107	390	1.17	0.270
MSSNG (Yuen2017)	ASD	1,625	232	1,188	484	1.17	0.298
Lelieveld2016	ID	820	211	661	211	1.32	0.257
DDD2017	DD	4,293	1,222	4,861	1,629	1.80	0.379

Similarly, we also gathered estimates of WES coverage and mutation rates from most of the clinical cohorts involved in this study. Our analysis shows that the exonic DNM rate of these cohorts is similar to the large-scale studies, ranging from 1.0-2.0 DNMs per exome (now Supplementary Table 5).

PatientID	cohort	cohort size	cohort WES coverage	overall DNM rate in the exome region	synonymous DNM rate in the exome region
NN1.p1, NN2.p1, NN3.p1	Nijmegen	4500	~120x	1.44	0.289
LG.p1	Leipzig	180	124x	~ 1 per trio	na
HU1.p1	BCM	8910	>100x	1-2 in trios	na
HU2.p1	BCM	8910	>100x	1-2 in trios	na
AN.p1	Amsterdam	277	~100x	~ 1 per trio	na
PF.p1	Paris	651	79x	~ 2 per trio	na
MA.p1	Melbourne	209	52x	0.97	0.39
SS1.p1, SS2.p1	SSC	2508	89.2x	1.08	0.256

Instead of excluding cohorts from our study, we determined how robust our significant excess of DNM would be to extremes of the DNM rate. Fortunately, the CH model incorporates the overall mutation rate as a custom parameter. We repeated the analysis using 1.0 ($P_{adj} = 8.4 \times 10^{-12}$) and 2.0 ($P_{adj} = 7.7 \times 10^{-9}$) as extremes of the overall mutation rate (we used 1.5 in the original analysis) in the CH model. We have incorporated these upper and lower bound estimates in the paper:

“Because exome sequence coverage can differ among centers and affect the sensitivity of *de novo* mutation discovery, we gathered empirical data regarding the rate of *de novo* mutations among the different referring centers (Supplementary Table 5). Instead of using 1.5 as the overall mutation rate in the CH model, for example, we repeated the analysis using 1.0 and 2.0 as extremes of the overall mutation rate in the CH model (Supplementary Table 5). The excess for *TANC2 de novo* LGD mutations among probands remains significant under conditions of the most extreme *de novo* mutation rate (*de novo*

mutation rate = 2.0, $P_{adj} = 7.7 \times 10^{-9}$; *de novo* mutation rate = 1.0, $P_{adj} = 8.4 \times 10^{-12}$) suggesting limited effect of differences in mutation detection sensitivity.”

- Analysis of negative control mutation type: previous studies often include synonymous mutations under the hypothesis that, as a class, synonymous mutations do not confer risk to disease (i.e., equal frequencies among cases and controls). Again, it would be reassuring to see a similar frequency of *de novo* synonymous mutations both between cohorts and between probands and siblings. If there is a significant difference, it would be better to either remove offending studies or normalize the data by the difference in the frequency of *de novo* synonymous mutations.

The table above shows that rates of synonymous mutation among studies are comparable and there is no difference of *de novo* synonymous mutation rate between SSC and ASC probands ($P = 0.25$, Poisson test). However, the *de novo* synonymous mutation rates between the SSC and MSSNG are different ($p < 2.2 \times 10^{-16}$, Poisson test). With the exception of the SSC, few studies have examined the rates of synonymous mutations in siblings. Although the overall mutation rate is slightly elevated for probands, almost all of this effect is due to an excess of severe *de novo* mutations (LGD and missense). In the SSC, there is no significant difference of *de novo* synonymous mutations between probands and siblings (see figure below).

Figure 1 | Rates of *de novo* events by mutational type in the SSC. Rates per child are estimated from the 403 joint coverage target region, then extrapolated for the entire exome. Mutation types are displayed by class, and the combined rate for all LGD mutations is shown at the bottom right. For each event type, the significance between probands and unaffected siblings is given. Sib, unaffected siblings. The error bars represent 95% confidence interval for the mean rates (Iossifov et al., Nature, 2014).

- It is worth noting, the MSSNG WGS cohort described in Yuen et al., Nature Neuroscience 2017 was sequenced on two different platforms, Illumina and Complete Genomics. The number of SNVs and indels appeared to differ significantly between technologies (Figure 2A and 2B of the manuscript) and thus the inclusion of this cohort may lower power for association due to the heterogeneous nature of the underlying data (as fewer variants were detected on the Complete Genomics platform).

We have excluded MSSNG WGS data from this study as suggested and now restrict our discovery analysis to two WES cohorts: SSC and ASC.

2) I must be missing something as I struggled to understand exactly how the gene prioritization methodology worked that culminated in 71 genes in Table 1. It came across that the only conditions were 1) that the gene was a target of RMRP or RBOX and 2) pass the pLI and RVIS cutoffs but then it wasn't clear how the *de novo* mutations (from SSC, ASC, and MSSNG) were involved.

The goal of this analysis was to prioritize novel autism candidate genes for consideration. Focusing on the *de novo* LGD mutations from the simplex quad families of **Simons Simplex Collection (SSC)**, we excluded all LGD mutations in genes that were recently implicated as candidates based on a combined analysis of WES from ~11,000 published samples with autism or DD (Coe et al., Nat Genet, 2019). We, thus, repeated the analysis excluding those genes. After that treatment we still observed a significant enrichment of *de novo* LGD mutations and RBFOX and FMRP binding genes in the SSC simplex quad families.

We, therefore, prioritized genes that were 1) FMRP or RBFOX binding, 2) intolerant to mutation (based on pLI and RVIS cutoffs), and 3) not identified previously as higher-likelihood candidates (Coe et al., Nat Genet, 2019) based on DNM data from three cohorts (**SSC, ASC and MSSNG**). There were 71 genes (now revised to 58 genes after removing samples sequenced by Complete Genomics in MSSNG) that met these criteria (Table 1). *De novo* mutation data of **ASC and MSSNG** cohorts was actually not involved in the statistical analysis, but only used for candidate genes prioritize based on the above three criteria.

We have revised the first and third paragraphs of the Results section to be clearer.

“Prioritizing novel ASD candidate genes. We reanalyzed WES data from 1,902 SSC simplex quad families with matched unaffected siblings as controls⁷. We investigated whether there is significant LGD mutation burden in probands compared to unaffected siblings after excluding genes with genome-wide *de novo* significance in a recent large-scale meta-analysis⁹. After removing common LGD mutations (minor allele frequency (MAF) > 0.1% present in the Exome Aggregation Consortium (ExAC) nonpsychiatric subset)^{16,17} and recurrent sites with low confidence, we annotated 307 *de novo* LGD mutations in probands (0.16 per individual) and 174 *de novo* LGD mutations in siblings (0.09 per individual) (Methods,

Supplementary Table 1). As expected, probands showed significant excess of *de novo* LGD mutations when compared to siblings ($P = 2.6 \times 10^{-9}$, ANOVA) (Methods and Fig. 1a). We repeated the analysis excluding variants in genes where genome-wide *de novo* significance had already been established⁹. The filtered burden analysis revealed about one third of the *de novo* LGD burden remained unaccounted (i.e., 0.13 LGD mutations per proband vs. 0.09 per sibling) ($P = 4.9 \times 10^{-4}$, ANOVA) (Fig. 1a).

Because ASD genes are significantly enriched as targets for FMRP and RBFOX binding^{2,4}, we further refined the filtered set of LGD candidates. We observed significant enrichment in FMRP ($P = 9.7 \times 10^{-9}$; OR = 2.2, Fisher's exact test) and RBFOX targets ($P = 2.2 \times 10^{-5}$; OR = 1.6, Fisher's exact test) (Fig. 1b) among probands but not siblings. We next assessed the distribution of each genome-wide significant gene's intolerance to mutation using the probability of being loss-of-function intolerant (pLI) score¹⁶ and the residual variation intolerance score (RVIS)¹⁸ as metrics (Supplementary Figure 1). Based on these distributions, we excluded LGD mutations in genes with pLI < 0.84 and RVIS percentiles > 32. Under such restrictions, the proband (1,902 SSC proband) burden for *de novo* LGD mutations became more significant (0.040 vs. 0.019; $P = 2.5 \times 10^{-4}$, ANOVA) (Fig. 1a).

To prioritize novel ASD candidate genes, we combined *de novo* LGD mutations from two main ASD family-based WES studies: namely, the SSC WES study and the Autism Sequencing Consortium (ASC) WES study⁴ (Methods). These two combined datasets represent 3,953 families. Based on the SSC simplex family analysis, we applied three criteria to prioritize the candidate genes: (i) mutation intolerance (pLI > 0.84 and RVIS% < 32), (ii) FMRP and RBFOX target enrichment and (iii) variants in genes where genome-wide *de novo* significance had not been established in the recent meta-analysis⁹. The procedure prioritized 58 ASD candidate genes for further consideration (Table 1, Fig. 1c, Supplementary Table 2)."

3) Combining published *de novo* mutations from WES and WGS with smMIP-sequencing but only correcting for 14 genes is statistically invalid, as Barrett et al., bioRxiv 2016 pointed out (<https://www.biorxiv.org/content/early/2017/07/07/115964>) when the same analysis strategy was done in Stessman et al., Nature Genetics 2016 (doi: 10.1038/ng.3792). As Barrett et al., pointed out, one can either: a) combine the data but correct for all genes in the genome or b) use only the smMIP-seq *de novo*s and correct for those genes in the second wave (i.e., 14 genes). Lastly, because the authors used two different mutational models for each of these 14 genes, they need to correct for each gene twice, not once.

We apologize for the confusion here. The referee is of course correct and we did not mean to imply that this was a significant finding and in fact explicitly stated that correction of this type would not withstand multiple test correction. Significance was not established by this analysis but rather the discovery of 9 (now revised to 10) *de novo* truncating mutations from 16,113 autism and developmental delay patients.

Assessing only for the frequency of *de novo* truncating mutations, we find a significant excess of *de novo* LGD mutations whether we use the chimpanzee-human model for divergence or the denovolyzeR model, which considers mutational context. This analysis survives multiple-text correction considering 18,946 genes in chimpanzee-human model and 19,618 in denovolyzeR model ($P_{adj} = 4.6 \times 10^{-10}$, CH model; $P_{adj} = 9.4 \times 10^{-10}$, denovolyzeR model).

However, we realized that we should not mention the correction data for the 14 genes, which is statistically invalid as the reviewer pointed out. We removed the p-values (correcting for 14 genes) and simply state that this enrichment would not survive multiple test correction.

“Combining the *de novo* mutations in *TANC2* identified in the SSC cohort (1 LGD and 1 missense), we observed a trend for an excess of LGD mutations in probands although this observation did not remain significant after genome-wide multiple-test correction.”

4) Could the authors explain the rationale for why those 14 specific genes were selected for smMIP-sequencing as opposed to any of the other 57 genes.

These 14 genes were selected from the larger pool of 71 genes based on practical considerations. We had developed a separate smMIP panel and we had nearly maximized the number of smMIP assays. The smMIP pool is sensitive to the total probe numbers so we selected the remaining 350 smMIPs based on the size of the gene and functional evidence (level of intolerance to mutation). We have added a note to the methods regarding the selection criteria.

4) It was unclear why the authors tested two different mutational models given that this imposes an additional testing correction and doesn't appear to provide any additional insight.

We chose to apply both models because they differ in underlying assumptions as well as the genes that are predicted (Coe et al., Nat Genet, 2019). DenovolyzeR, for example, considers mutational context and longer evolutionary distances to estimate DNM rate (human-macaque) while the CH model considers shorter evolutionary distances (chimpanzee-human). While both perform similarly, there are a subset of genes that are detected by one gene and not the other. In the case of *TANC2*, both models predicted an excess of LGD DNMs. The reviewer is correct, however, in that we did not consider the additional testing correction and have done so in the revised manuscript (correcting twice for the use of two models and ~19,000 gene models).

“We estimate a significant excess for *TANC2 de novo* LGD mutations compared with expected calculations ($P = 2.4 \times 10^{-14}$, CH model; $P = 4.8 \times 10^{-14}$, denovolyzeR model) that remained significant after genome-wide multiple-test correction ($P_{adj} = 9.2 \times 10^{-10}$, CH model; $P_{adj} = 1.9 \times 10^{-9}$, denovolyzeR model, Bonferroni correction for two models and ~19000 genes).”

It would be appreciated if the authors would add the following citations:

- pLI (Lek et al., Nature 2016) (line 117)
- RVIS (Petrovski et al., PLOS Genetics 2013) (line 117)
- ExAC was used by not cited (line 103) (Lek et al., Nature 2016)
- Filtering De novo mutations using ExAC should be cited (Kosmicki et al., Nature Genetics 2017)
- 1000 Genomes Project (line 160)
- ESP6500 (line 160)

We have added the citations as suggested.

Minor comments:

- On line 180 and 181, I assume instead of another five case had no autistic phenotypes the authors meant the remaining five cases had no autistic phenotypes? Also, it should be cases not case.

We have revised them accordingly.

“Of the 20 individuals assessed for ASD, seven cases had a formal ASD diagnosis, eight cases had clinical impression of ASD or Rett-like features, while remaining five cases had no recognized autism features.”

- Could the authors add confidence intervals to Figures 1a and 1b?

We added the confidence intervals to Figure 1a. The bar in Figure 1b represents the $-\log_{10}(\text{p-values})$, which is not applied for confidence intervals.

Fig. 1 Prioritizing ASD candidate genes based on gene intolerance metrics and enrichment of FMRP/RBFOX targets. a Burden of *de novo* LGD mutations in probands of SSC simplex quad families for three categories: i) all *de novo* LGD events; ii) *de novo* LGD events excluding genes where significant burden has been reached; and iii) *de novo* LGD mutations in intolerant genes without significance. The errors bar represents a 95% confidence interval for the mean rates. b Enrichment of genes with *de novo* LGD mutations in FMRP and RBFOX targets. Enrichment is performed after excluding genes that reached significance (top panel) and the same but requiring that the genes are intolerant to mutation (bottom panel). c Selection of genes for targeted sequencing. 128 intolerant genes were prioritized by pLI score and RVIS percentile (pLI < 0.84 and RVIS percentiles > 32) from the SSC and ASC cohorts of which 58 genes were FMRP/RBFOX targets (Table 1). 14 genes were selected from the 58 genes for targeted sequencing in 2,514 ASD probands from the ACGC cohort.

- Which cohorts comprise the 8 cohorts specified on lines 164-165?

The 8 cohorts (now revised to 9 cohorts) were presented in Table 2. To make it clearer, we have added the name of the 9 cohorts in the main text.

“Ten patients with confirmed *TANC2* *de novo* LGD mutations were screened from nine cohorts from a total of 16,113 individuals (ACGC, BCM, Leiden, Leipzig, Melbourne, Paris, SSC, Toronto, Table 2).”

Reviewer #2 (Remarks to the Author):

In this manuscript, Guo et al. undertake a large scale, systematic study to identify and characterize human mutations in the gene *TANC2* in neurodevelopmental disorders. The sheer amount of experimental and logistical work involved in this human disease project is impressive. In particular, they provide both genetic and clinical data for 20 individuals identified with mutations in *TANC2* and provide evidence that these mutations are associated with a spectrum of neurological and neuropsychiatric diseases, including intellectual disability, autism spectrum disorder, epilepsy, and others. Finally, because *TANC2* interacts with components at the postsynaptic density, they go on to study *Drosophila* neuromuscular junctions. Using neuronal RNAi knockdown of the fly homolog of *TANC2*, *rols*, the authors report a moderate increase in synaptic growth. Together, they conclude disruption of *TANC2* contributes to a variety of seemingly disparate neurodevelopmental diseases.

Although *TANC2* has been implicated in neurodevelopmental diseases in humans previously, the identification of new human disease-causing alleles of *TANC2* and their clinical description are important. The PSD is well known to be a key target for many neurodevelopmental, neurological, and psychiatric diseases, and the identification of specific alleles of *TANC2* and their clinical manifestations is exciting. However, the characterization of *rols* in *Drosophila* is very preliminary and needs major additional work. In particular, several key genetic experiments, interpretations, and additional assays need to be performed to properly assess the role of *rols* at synapses, as well as to validate and extend their insights. Without these experiments, there is little additional insight into how *TANC2/Rols* may impact synaptic development and/or function.

We have worked extensively over the last five months to further clarify the role of the *TANC2* ortholog *rols* in neurodevelopment in *Drosophila*. We specifically teamed up with an expert in this area, Hugo Bellen, who working with his students performed a series of knockdown/knockout and rescue experiments for *rols*. The results are now presented as three new figures (Figs. 5-7) and a much more expanded Results section. Our revised analysis confirms a role of *TANC2 (rols)* in postsynaptic function as well as the glia function in *Drosophila*.

We generated new *rols* alleles that allow visualization of its expression and carried out a series of expression and phenotype analyses along with tissue-specific RNAi-mediated knockdown studies. Unlike humans where *TANC2* is expressed extensively in radial glial and neurons, we did not observe *rols* in most neuronal cells, including the motor neurons. However, we observed that *rols* was present in muscle cells earlier on in development with its proposed roles in early muscle development. As a result, we did not investigate further the function of *rols* in presynapse in the revised manuscript. Instead, as the reviewer suggested, we investigated whether reduction of *rols* influences synapse growth and function in

postsynaptic membranes independent of its role in myoblast fusion during embryogenesis, we carried out RNAi experiments using muscle-specific driver. Interestingly, our manipulation did not affect early muscle development, likely due to the specific GAL4 lines used in this work (*C57-GAL4*). We observed a significant increase in mean satellite bouton number and significant reduction of the synaptic immunoreactivity of GluRIIA in the muscle-specific knockdown lines. Considering these phenotypes were rescued by co-expression of wild-type *rols* using a *UAS-rols* transgene, these data suggest that in addition to the known role of *rols* in early muscle development during embryogenesis, this gene also plays a role later in muscles to establish a proper neuromuscular junction.

In addition to identifying an evolutionarily conserved role for *rols* in post-synaptic cells, we observed enriched expression of *rols* in both larva and adult glia cells. To further assess whether *rols* expressed in glia cells are necessary for neural function, we conducted glial-specific (*repo-Gal4*) knockdown of *rols* and assess adult male flies for any observable changes in behavior. Interestingly, we observed that glial-specific *rols* knockdown flies displayed an increased total duration of wing-extension courtship behavior while maintaining wild-type levels of locomotion, grooming, and copulation. To our knowledge, this is the first time that a *rols* defect in glial has been shown to be required for normal behavior in any species. While the significance of the *Drosophila* model to the human phenotype is not clear, it is interesting that both human homologs of *rols* show a broader expression profile in both radial glial and neuronal cells, although TANC1 is more highly expressed in radial glial. This raises the exciting possibility that radial glial defects may be contributing to the human phenotype.

Major points:

1. One major question the authors fail to consider is that TANC2/Rols is a PSD interacting protein with clear roles in dendrites (the authors cite the relevant papers here and underscore this point in the abstract). However, at the fly NMJ, the muscle is the postsynaptic compartment with high expression of PSD95 (DLG) and other postsynaptic components. However, the authors reduce *rols* expression in presynaptic neurons (both pan-neuronal and motor-neuron RNAi knockdown). How do the authors interpret a presynaptic function of *rols* in regulating synaptic growth (through an undermined mechanism) with the established functions of TANC2 at the PSD? This seems to be a major conundrum that is not discussed or considered.

The reviewer raises a fair point. In the revised analysis, we investigated whether reduction of *rols* influences synapse growth and function in postsynapse. As we stated above, we observe a significant

increase in mean satellite bouton number and significant reduction of the synaptic immunoreactivity of GluRIIA in the muscle-specific knockdown lines (Fig. 6).

Fig. 6 *rols* is historically expressed in developing muscle cells and muscle-specific knockdown of *rols* increases satellite bouton number and decreases GluRIIA levels at the 3rd instar larval NMJ. **a** G-TRACE analysis by crossing *rols*^{MI02479-TG4.1} to UAS-RedStinger, UAS-FLP, Ubi-p63E(FRT.STOP)Stinger reveals that *rols* is currently expressed in wrapping glia at the NMJ (yellow arrow) and was historically expressed in muscle cells (white arrows). Scale bar = 50 μ m. **b-c** muscle-specific (*C57-Gal4*) knockdown of *rols*

showed an increased in number of satellite boutons (b) and decreased normalized fluorescent intensity of GluR IIA (c) compared to wild-type larvae. Scale bar = 10 μ m. d Statistical results for b-c. **** p < 0.00001, ***p < 0.0001, **P < 0.01, *P < 0.05; n.s., not significant. Error bars represent SEM.

2. There are several important controls in assessing synaptic growth at the NMJ and the function of rols that should be performed. Fortunately, these experiments should be relatively straightforward. First, there are 2 rols RNAi lines available from the TRiP library at the Bloomington *Drosophila* Stock Center. It is not clear from the methods whether the single RNAi line they used is one of these, but regardless the second RNAi lines as well as additional neuronal drivers (nsyb-Gal4, D42-Gal4, etc) can be used. In addition, because TANC2 is a PSD-interacting protein, and many studies on fly rols have demonstrated functions in the postsynaptic muscle, postsynaptic RNAi of rols (using G14-Gal4, MHC-Gal4, etc) should be performed and synaptic growth assessed. More importantly, a variety of loss-of-function rols alleles (ex18, etc) and UAS-rols transgenic lines have been generated and are available. These genetic reagents can be used to assess NMJs of rols mutants and to rescue any phenotypes observed by expression of rols in neurons or muscle.

As suggested, we obtained two *rols* RNAi lines and one transgenic line (see table below) from the TRiP library at the Bloomington *Drosophila* Stock Center. Using a muscle-specific driver (C57-Gal4), we knocked-down *rols* in muscle via RNAi and examined the 3rd instar NMJ. We observed a significant increase in mean satellite bouton number and significantly reduction of the synaptic immunoreactivity of GluRIIA in the muscle-specific knockdown lines.

No.	Catalog	Genotype	Stock Center
UAS-RNAi-1	56986	$y^1 sc^* v^1$; P{TRiP.HMC04426}attP	Bloomington
UAS-RNAi-2	58262	$y^1 sc^* v^1$; P{TRiP.HMJ22326}attP40	Bloomington
UAS-OE	22194	$y^1 w^*$; P{Mae-UAS.6.11}rolsLA00796	Bloomington

3. There are a variety of additional experiments that could be performed to gain insight into rols function at the NMJ. Ideally, electrophysiological recordings would be performed in rols mutants/RNAi lines to assess impacts on synaptic function, although this Reviewer understands this approach may be beyond the scope and toolkit of the authors. However, there appears to be reagents available from public stock centers as well as from other investigators to assess rols expression (rols promoter-Gal4 fusions), transgenic overexpression/rescue lines (UAS-rols as well as UAS-TANC2), as well as perhaps tagged constructs and/or antibodies. These could be used to reveal the subcellular localization of Rols at the NMJ (at the PSD in muscle and/or in motor neurons) and to assess whether overexpression of rols impacts

synaptic growth (perhaps inhibiting growth?). These experiments could provide more insight into how disruption of *rols* impacts synaptic growth or function. At minimum, additional immunostaining against presynaptic active zones and postsynaptic glutamate receptors using established antibodies can be performed to better detail NMJ synapses in *rols* mutants/RNAi lines. Since *rols* is a putative PSD gene, was the organization of DLG altered at the PSD in *rols* RNAi? Or mutants?

In the revised manuscript, we performed a series of expression analyses of *rols* in *Drosophila*. Unfortunately, we were unable to detect *rols* in most neuronal cells, including motor neurons. However, we observed that *rols* was present in muscle cells earlier during development. So, we assessed synapse growth and function in muscle-specific RNAi lines. We observed a significant increase in mean satellite bouton number and significant reduction of the synaptic immunoreactivity of GluRIIA in the muscle-specific knockdown lines. These data indicate that *rols* in the post-synapse is important for the normal development of the NMJ.

Minor point:

1. Typo on line 196

Corrected.

Reviewer #3 (Remarks to the Author):

This paper describes what was initially a refreshingly simple approach to plausible developmental disease gene identification. This begins with the Simons Simplex quad whole exome data. The authors have compared the *de novo* likely gene disruptive events on a per gene and combined basis so confirm the excess of LGD events in the probands remains when the known genes are excluded. The then filtered the genes by their reported association as mRNA with FMRP etc to show further enrichment. The candidate genes were selected as high pLi and RVIS levels chosen on the basis of a 1 in 20 false discovery rate. This is all sensible and standard approach. This approach became significantly less easy to assess and it was then repeated using a combined cohort including two groups with which I am unfamiliar, particularly with the case mix or repertoire of known causative gene *de novo* mutations in ASC and MSSNG. This led to 71 genes being targeted for further analysis (many of which would be considered known neurodevelopment disorder genes).

This point also confused the first reviewer and we realized that we needed to make this description clearer. The goal of this analysis was to prioritize novel autism candidate genes for consideration. Focusing on the *de novo* LGD mutations from the Simons Simplex Collection (SSC), MSSNG and ASC, we excluded all LGD mutations in genes that were recently implicated as candidates based on a combined analysis of WES from ~11,000 published samples with autism or DD (Coe et al., Nat Genet, 2019). We then repeated the aggregate analysis for *de novo* mutation burden after excluding those genes. The analysis shows that we still observe a significant enrichment of *de novo* LGD mutations and RBFOX and FMRP binding genes with *de novo* mutations in probands when compared to siblings. We, therefore, prioritized genes that were 1) FMRP or RBFOX binding, 2) intolerant to mutation (based on PLI and RVIS cutoffs), and 3) not identified previously as higher-likelihood candidates (Coe et al., Nat Genet, 2019). There were 71 genes (now revised to 58 genes due to the request of the first reviewer to exclude MSSNG) that met these criteria (Table 1).

We have revised the first paragraph of the Results section and the third paragraph of the discussion to be clearer:

“Prioritizing novel ASD candidate genes. We reanalyzed WES data from 1,902 SSC simplex quad families with matched unaffected siblings as controls⁷. We investigated whether there is significant LGD mutation burden in probands compared to unaffected siblings after excluding genes with genome-wide *de novo* significance in a recent large-scale meta-analysis⁹. After removing common LGD mutations (minor allele frequency (MAF) > 0.1% present in the Exome Aggregation Consortium (ExAC) nonpsychiatric subset)^{16,17} and recurrent sites with low confidence, we annotated 307 *de novo* LGD mutations in probands (0.16 per individual) and 174 *de novo* LGD mutations in siblings (0.09 per individual) (Methods,

Supplementary Table 1). As expected, probands showed significant excess of *de novo* LGD mutations when compared to siblings ($P = 2.6 \times 10^{-9}$, ANOVA) (Methods and Fig. 1a). We repeated the analysis excluding variants in genes where genome-wide *de novo* significance had already been established⁹. The filtered burden analysis revealed about one third of the *de novo* LGD burden remained unaccounted (i.e., 0.13 LGD mutations per proband vs. 0.09 per sibling) ($P = 4.9 \times 10^{-4}$, ANOVA) (Fig. 1a).

Because ASD genes are significantly enriched as targets for FMRP and RBFOX binding^{2,4}, we further refined the filtered set of LGD candidates. We observed significant enrichment in FMRP ($P = 9.7 \times 10^{-9}$; OR = 2.2, Fisher's exact test) and RBFOX targets ($P = 2.2 \times 10^{-5}$; OR = 1.6, Fisher's exact test) (Fig. 1b) among probands but not siblings. We next assessed the distribution of each genome-wide significant gene's intolerance to mutation using the probability of being loss-of-function intolerant (pLI) score¹⁶ and the residual variation intolerance score (RVIS)¹⁸ as metrics (Supplementary Figure 1). Based on these distributions, we excluded LGD mutations in genes with pLI < 0.84 and RVIS percentiles > 32. Under such restrictions, the proband (1,902 SSC proband) burden for *de novo* LGD mutations became more significant (0.040 vs. 0.019; $P = 2.5 \times 10^{-4}$, ANOVA) (Fig. 1a).

To prioritize novel ASD candidate genes, we combined *de novo* LGD mutations from two main ASD family-based WES studies: namely, the SSC WES study and the Autism Sequencing Consortium (ASC) WES study⁴ (Methods). These two combined datasets represent 3,953 families. Based on the SSC simplex family analysis, we applied three criteria to prioritize the candidate genes: (i) mutation intolerance (pLI > 0.84 and RVIS% < 32), (ii) FMRP and RBFOX target enrichment and (iii) variants in genes where genome-wide *de novo* significance had not been established in the recent meta-analysis⁹. The procedure prioritized 58 ASD candidate genes for further consideration (Table 1, Fig. 1c, Supplementary Table 2).”

The paper then shifts to concentrate on *TANC2* in which only one LGD DNM had been identified in SSC to sequencing of a further cohort and to expand the analysis to include DNM that are NSV. Matchmaker was used to assemble a total of 19 families which includes 3 intragenic CNV. 11 of the LGD events were *de novo*, one inherited from a mosaic parent. In the other inherited cases are taken as evidence for a strong psychiatric predisposition associated with carriers of LGD events.

This is an interesting paper and *TANC2* may indeed represent a genuinely causative NDD gene. However I cannot judge on the basis of the presented data how likely this is to be due to ascertainment bias alone. The phenotypic information is not compelling as there is no indication of how unusual the "pattern" is compared to randomly selected groups of 20 cases from the myriad of cohorts represented in the mutation table.

To address this question, we determined the number of patients that were represented by each referring clinical lab. We observe a significant enrichment of *de novo* mutations in *TANC2* even after adjusting for the total number of genes in the genome. We identified 9 (now revised to 10) *de novo* truncating mutations from 16,113 autism and developmental delay patients. Assessing only for the frequency of *de novo* truncating mutations, we find a significant excess of *de novo* LGD mutations whether we use the chimpanzee-human model for divergence or the denovolyzeR model, which considers mutational context. This analysis survives multiple-text correction considering 18,946 genes in chimpanzee-human model and 19,618 in denovolyzeR model ($P_{\text{adj}} = 4.6 \times 10^{-10}$, CH model; $P_{\text{adj}} = 9.4 \times 10^{-10}$, denovolyzeR model, Bonferroni correction). This finding is also robust to differences in exome sequence coverage and differences in DNM sensitivity.

As a second test, we also performed a more traditional burden analysis between cases and controls (Fisher's exact test). For example, we identified 13 probands carrying LGD mutations with a known cohort size ($n = 17,567$), which represents a nominally significant enrichment of LGD mutations in NDD patients when compared to 45,375 ExAC nonpsychiatric control samples. To guard against potential exon dropout, we assessed the mean coverage of *TANC2* exons in ExAC. The average coverage is 49.2 sequence reads per exon (Supplementary Fig. 3) with 24/25 of the exons showing on average more than 20-fold sequence read coverage.

In addition, there are four other observations that support pathogenicity of the disruptive mutations:

1. We have four autosome dominant families with inherited LGD mutations. The mutation co-segregates with neurodevelopmental phenotypes in the families (Fig. 2C). Importantly, families NN2 and CF have multiple affected siblings.
2. We also identified three individuals with intragenic CNVs in families with neurodevelopmental delay. Such CNVs have not been observed in controls.
3. We found that disruption of *rols* (*TANC2*) interferes with synapse growth and function in *Drosophila* postsynapse.
4. In addition, one of the loss-of-function mutations (p.R1066*) in the SSC patient has been investigated functionally and the mutant protein failed to accumulate at the dendritic spines (Stucchi et al., Cell Rep, 2018). We have mentioned this observation and included the citation in the manuscript.

The reviewer is correct; the phenotype here is not strikingly syndromic but rather the children have a relatively general pattern of autism and developmental delay. There are, however, some features that are

enriched. For example, epilepsy, which is not common in most of the cohorts, is reported in over 50% of our patients. In addition, we observed the potential distinctive facial features (Fig. 3) but larger sample sizes of patients will be required to confirm.

I accept that the lack of LGD events in ExAC and gnomAD is interesting but there are 8 LGD events as this is a highly biased sample which has been selected for the presence of a LGD DNM in *TANC2*. The main deficiency is the lack of a very large scale cohort that is genuinely unselected for disease.

We agree that there is potential biased sampling since the patients are selected from the cohorts with *TANC2* LGD mutations. We attempted to mitigate for platform & ascertainment biases. We focused, for example, exclusively ExAC as opposed to gnomAD because of differences in platform sensitivity (exome versus genome) and the fact that there was a well-defined nonpsychiatric comparison set to use as controls (Note: when you limit the analysis to those that are nonpsychiatric controls the number of LGD events in ExAC drops from six to three which is also suggestive of an enrichment of LGD in psychiatric samples). To guard against potential exon dropout, we assessed the mean coverage of *TANC2* exons in ExAC. The average coverage is 49.2 sequence reads per exon (Supplementary Fig. 3) with 24/25 of the exons showing on average more than 20-fold sequence read coverage. While ExAC is not ideal, it has the best set of controls to which we have access. It should be noted, however, that significance is determined based on the excess of LGD *de novo* mutations, which is independent of the population controls.

It would be useful to have some idea of the total number of cases screened to generate these 19 families - and the distribution of clinical features. However I accept that this is an extremely difficult task so my preference would be for a more circumspect approach to the assignment of pathogenesis.

We have gathered that data and include it in Table 2. There are 13 cases with *TANC2* LGD mutations screened from 11 cohorts with known sample size of 17,567. The case-control analysis between those 17,567 cases and the ExAC non-neuropsychiatric samples revealed 11.2-fold enrichment in cases ($P = 1.43 \times 10^{-5}$) although this does not yet survive genome-wide multiple testing ($P_{adj} = 0.29$, Bonferroni correction for 20,000 genes). The other patients with *TANC2* LGD mutations are from the individual clinical settings, of which, unfortunately, the sample size cannot be determined.

What we know for the clinical features of those cohorts is the primary diagnosis. We have added it as a “primary diagnosis” column in the revised Table 2. Unfortunately, the distribution of co-occurring conditions could not be obtained prospectively for most of the cohorts.

Based on the referee’s suggestion we have attempted to be more circumspect in our concluding remarks and have revised as follows:

“In summary, we have demonstrated an excess of *de novo* disruptive *TANC2* mutations among patients with neurodevelopmental delay and autism. Our results suggest that *TANC2* is one of a growing list of genes where mutations associate with both pediatric neurodevelopmental and adult neuropsychiatric disease, for example *SHANK3*⁴⁰, *NRXN1*⁴¹, *ZMYND11*⁴², *POGZ*²⁵, and *RELN*⁴³. Because disruptive mutations may also be inherited, much larger cohorts of both cases and controls will be needed to establish its significance and involvement with neurodevelopment and psychiatric disease. As such, the detection of pathogenic *TANC2* mutations will not only increase the diagnostic yield in such populations but also be important for a better understanding of genotype-phenotype associations and guiding precision medicine. Animal models and deeper molecular functional studies on the cell-specific role of *TANC2* in neurodevelopment will not only provide insights regarding PSD pathogenesis in ASD and other neuropsychiatric disorders but also provide an avenue for the development of long-term treatment of these disorders.”

From the pathogenesis assessment perspective, our *rols* (*TANC2* homologs in *Drosophila*) RNAi *Drosophila* model showed the abnormal synapse growth. In this revised manuscript, we carried out extensive expression and phenotype analyses along with tissue-specific RNAi-mediated knockdown on *Drosophila*.

Reviewers' Comments:

Reviewer #1:

Remarks to the Author:

I thank the authors for addressing all of my concerns, and I feel the manuscript now warrants publication in Nature Communications.

Reviewer #2:

Remarks to the Author:

In this revised manuscript, Guo et al. have bolstered and clarified a variety of issues related to their statistical analysis of ASD studies and their decision to focus on TANC2 for follow up studies. In addition, they have focused on generating a host of new reagents to more rigorously characterize the expression and functions of the *Drosophila* homolog of TANC2, *rols*, in postsynaptic structure and, surprisingly, glia. In particular, they developed and incorporated many new genetic approaches to reveal that in *Drosophila*, *rols* is surprisingly not expressed in most neurons and instead is highly expressed in glia, with early expression in muscle as well. Although diverging in important ways when compared the results and interpretations in their original manuscript, these new genetic resources and findings will provide an important foundation for the field to resolve the mechanistic functions of *rols* in postsynaptic compartments and glia. They also provide some evidence for a glial-specific function of *rols* in influencing mating behavior. There are several questions that remain unresolved despite all of this new data, but I recognize the major effort that was required to get this far in the limited time allowed for the revision and am ok letting the field work out the answers in future studies. While it would have been great to achieve more conclusive insights into postsynaptic and glial functions of *rols* in *Drosophila*, the authors have largely responded to my key concerns in this improved revision. I do, however, have a few concerns for the authors to consider.

1. Clearly *rols* is an essential gene and the use of two independent RNAi lines to characterize the role of *rols* in muscle and glia make sense. While it seems strange that *rols* is not expressed in muscle throughout larval stages, perdurance of the protein from earlier stages and/or developmental contributions to PSD organization could give rise to the reduction in GluRIIA levels reported. Nonetheless, the authors now report "satellite" boutons formed on presynaptic boutons due to a postsynaptic function of *rols*. To my knowledge, it is usually presynaptic processes that lead to this satellite bouton formation, usually defects in neuronal endocytosis (see Dickman et al., *Current Biology*, 2006, others on endophilin mutants, synaptojanin, *dap160*, nervous wreck, etc). Although unusual to have a postsynaptic gene involved in this phenotype, there is one report of *diablo/Henji* regulating postsynaptic glutamate receptor abundance and satellite boutons (Wang et al., *PLoS Genetics*, 2016). The authors may consider raising these points in their discussion.

2. In the original manuscript, the authors reported neuronal RNAi of *rols* to enhance synaptic growth, while in this revised manuscript they show *rols* is apparently not even expressed in neurons. Were the previous results just artifacts of genetic background or off-target effects?

3. minor point: line 333 – GluRIIA is a glutamate receptor subunit, not a receptor type. The authors should refer to "GluRIIA-containing receptors" or "GluRIIA receptor subunit levels" in their manuscript to be clear about this.

Signed,

Dion Dickman

Reviewer #3:

Remarks to the Author:

The authors have provided a very comprehensive response to the reviewers comments and the revised manuscript is considerably improved. I have residual concerns that the null hypothesis is not front and centre in the conclusions drawn in the abstract and that the problems associated with the complexity of the data and analytical history is being under-appreciated. Recognising bias associated with the ascertainment is key. For example no statistical analysis supporting the neuropsychiatric phenomena being specifically associated with TANC2 is provided yet this is highlighted in the abstract.

I did appreciate the inclusion of the Fishers exact testing against the ExAC data. This does not show a significant enrichment in TANC2 LRG when corrected for multiple testing. However the other analysis does provide reasonable support an associated on de novo events in neurodevelopmental disease although with very broad phenotypic consequences.

I remain confused about the status of this locus as a causative gene but it is reasonable to present the data to the scientific community. My only concern is the significant uncertainties are not being shared in the abstract - which is all many people will read.

Minor points

Table 2: add the genome build to the column header

REVIEWERS' COMMENTS:

Reviewer #1 (Remarks to the Author):

I thank the authors for addressing all of my concerns, and I feel the manuscript now warrants publication in Nature Communications.

Thank you.

Reviewer #2 (Remarks to the Author):

In this revised manuscript, Guo et al. have bolstered and clarified a variety of issues related to their statistical analysis of ASD studies and their decision to focus on TANC2 for follow up studies. In addition, they have focused on generating a host of new reagents to more rigorously characterize the expression and functions of the *Drosophila* homolog of TANC2, *rols*, in postsynaptic structure and, surprisingly, glia. In particular, they developed and incorporated many new genetic approaches to reveal that in *Drosophila*, *rols* is surprisingly not expressed in most neurons and instead is highly expressed in glia, with early expression in muscle as well. Although diverging in important ways when compared the results and interpretations in their original manuscript, these new genetic resources and findings will provide an important foundation for the field to resolve the mechanistic functions of *rols* in postsynaptic compartments and glia. They also provide some evidence for a glial-specific function of *rols* in influencing mating behavior. There are several questions that remain unresolved despite all of this new data, but I recognize the major effort that was required to get this far in the limited time allowed for the revision and am ok letting the field work out the answers in future studies. While it would have been great to achieve more conclusive insights into postsynaptic and glial functions of *rols* in *Drosophila*, the authors have largely responded to my key concerns in this improved revision. I do, however, have a few concerns for the authors to consider.

1. Clearly *rols* is an essential gene and the use of two independent RNAi lines to characterize the role of *rols* in muscle and glia make sense. While it seems strange that *rols* is not expressed in muscle throughout larval stages, perdurance of the protein from earlier stages and/or developmental contributions to PSD organization could give rise to the reduction in GluRIIA levels reported. Nonetheless, the authors now report “satellite” boutons formed on presynaptic boutons due to a postsynaptic function of *rols*. To my knowledge, it is usually presynaptic processes that lead to this satellite bouton formation, usually defects in neuronal endocytosis (see Dickman et al., *Current Biology*, 2006, others on endophilin mutants, synaptojanin, *dap160*, nervous wreck, etc). Although unusual to have a postsynaptic gene involved in this phenotype, there is one report of *diablo*/*Henji* regulating postsynaptic glutamate receptor abundance and

satellite boutons (Wang et al., PLoS Genetics, 2016). The authors may consider raising these points in their discussion.

We thank the reviewer for the suggesting these points. We have added a paragraph to the discussion highlighting these aspects.

“In *Drosophila*, a reduction of *rols* in muscles results in reduction of post-synaptic GluRIIA levels. However, we only detected *rols* expression in muscle early in development and not throughout larval developmental stages. The effects of *rols* knockdown in embryonic muscle using C57-GAL4 may affect NMJ development or the perdurance of *Rols* in larval muscles may affect the organization of the PSD and cause a decrease in GluRIIA levels when *Rols* is lost progressively. We also observed an increase in presynaptic satellite bouton number upon reduction of *rols* in muscle. Typically, satellite bouton formation is determined by presynaptic processes usually associated with defects in synaptic vesicle endocytosis that lead to an increase in satellite boutons³⁹⁻⁴¹. In contrast, knockdown of the post-synaptic protein *Diablo* in muscle results in a decrease in satellite bouton number⁴². The latter is correlated with an increase in GluRIIA levels. Hence, the mechanisms underlying these NMJ defects when *rols* is affected remain to be determined.”

2. In the original manuscript, the authors reported neuronal RNAi of *rols* to enhance synaptic growth, while in this revised manuscript they show *rols* is apparently not even expression in neurons. Were the previous results just artifacts of genetic background or off-target effects?

The referee is correct that in the original manuscript we did observe some evidence for a synapse overgrowth phenotype after neuronal knockdown of *rols*. Because the effect was relatively modest and *Drosophila* functional studies are not a strong focus of research in our lab, we solicited the assistance of the Bellen lab, world-renowned experts in *Drosophila* neuronal functional studies. Using the new reagents developed, we did not observe significant expression of *rols* in most neurons. However, we cannot exclude the possibility that *rols* may be expressed in a small subset or the potential of off-target effects as suggested. Because the glial expression signal was so much stronger, after considerable discussion, we decided not to include the neuronal phenotypes of the neuronal RNAi lines in the revised manuscript but rather focus on the glial work. The referee’s comments were critical in this assessment.

3. minor point: line 333 – GluRIIA is a glutamate receptor subunit, not a receptor type. The authors should refer to “GluRIIA-containing receptors” or “GluRIIA receptor subunit levels” in their manuscript to be clear about this.

We have changed “type-A Glutamate receptor” to “GluRIIA receptor subunit levels”.

Reviewer #3 (Remarks to the Author):

The authors have provided a very comprehensive response to the reviewers comments and the revised manuscript is considerably improved. I have residual concerns that the null hypothesis is not front and centre in the conclusions drawn in the abstract and that the problems associated with the complexity of the data and analytical history is being under-appreciated. Recognising bias associated with the ascertainment is key. For example no statistical analysis supporting the neuropsychiatric phenomena being specifically associated with TANC2 is provided yet this is highlighted in the abstract.

We have revised the abstract highlighting the statistical uncertainty and the need to replicate the neuropsychiatric aspect.

“Although this observation requires replication to establish statistical significance, it also suggests that mutations in this gene are associated with a variety of neuropsychiatric disorders consistent with its postsynaptic function”

I did appreciate the inclusion of the Fishers exact testing against the ExAC data. This does not show a significant enrichment in TANC2 LGD when corrected for multiple testing. However the other analysis does provide reasonable support an associated on de novo events in neurodevelopmental disease although with very broad phenotypic consequences.

I remain confused about the status of this locus as a causative gene but it is reasonable to present the data to the scientific community. My only concern is the significant uncertainties are not being shared in the abstract - which is all many people will read.

We have revised the abstract highlighting the statistical uncertainty and the need to replicate the neuropsychiatric aspect.

Minor points

Table 2: add the genome build to the column header

Added.